
**Transient conduit permeability controlled by a shift between compactant shear and dilatant**
**rupture at Unzen volcano (Japan)**
Lavallée Yan[1*], Miwa Takahiro[2], Ashworth James D.[1], Wallace Paul A.[1,3], Kendrick Jackie E.[1,4], Coats
Rebecca[1], Lamur Anthony[1], Hornby Adrian[5], Hess Kai-Uwe.[6], Matsushima Takeshi[7], Nakada Setsuya[8],
Shimizu Hiroshi[8], Ruthensteiner Bernhard[9], Tuffen Hugh[10]
[1] Earth, Ocean and Ecological Sciences, University of Liverpool, Liverpool, United Kingdom
[2] Earthquake Research Department, National Research Institute for Earth Science and Disaster
Resilience (NIED), Tsukuba, Japan
[3] Department of Geosciences, Environment and Society, Université Libre de Bruxelles, Brussels,
Belgium
[4] Geosciences, University of Edinburgh, Edinburgh, United Kingdom
[5] Earth and Atmospheric Sciences, Cornell University, United States of America
[6] Earth and Environmental Sciences, Ludwig-Maximilians University of Munich, Germany
[7] Institute of Seismology and Volcanology, Faculty of Sciences, Kyushu University, Shimabara,
Nagasaki, Japan
[8] National Research Institute for Earth Science and Disaster Resilience, Tennodai, Tsukuba, 305-
0006, Japan
[9] Staatliche Naturwissenschaftliche Sammlungen Bayerns (SNSB), Zoologische Staatssammlung
München, München, Germany.
[10] Earth Sciences, University of Lancaster, United Kingdom
*ylava@liverpool.ac.uk
**ABSTRACT**
The permeability of magma in shallow volcanic conduits controls the fluid flow and pore pressure
development that regulates gas emissions and the style of volcanic eruptions. The architecture of the
permeable porous structure is subject to changes as magma deforms and outgasses during ascent. Here,
we present a high-resolution study of the permeability distribution across two conduit shear zones
(marginal and central) developed in the dacitic spine that extruded towards the closing stages of the
1991-1995 eruption at Unzen volcano, Japan. The marginal shear zone is approximately 3.2 m wide
and exhibits a 2-m wide, moderate shear zone with porosity and permeability similar to the conduit
core, transitioning into a ~1-m wide, highly-sheared region with relatively low porosity and
permeability, and an outer 20-cm wide cataclastic fault zone. The low porosity, highly-sheared rock
further exhibits an anisotropic permeability network with slightly higher permeability along the shear
plane (parallel to the conduit margin) and is locally overprinted by oblique dilational Riedel fractures.
The central shear zone is defined by a 3-m long by ~9-cm wide fracture ending bluntly and bordered



by a 15-40 cm wide damage zone with an increased permeability of ~3 orders of magnitude; directional
permeability and resultant anisotropy could not be measured from this exposure.
We interpret the permeability and porosity of the marginal shear zone to reflect the evolution of
compactional (i.e., ductile) shear during ascent up to the point of rupture, estimated by Umakoshi et al.
(2008), at ~500 m depth. At this point the compactional shear zone would have been locally overprinted
by brittle rupture, promoting the development of a shear fault and dilational Riedel fractures during
repeating phases of increased magma ascent rate, enhancing anisotropic permeability that channels fluid
flow into, and along, the conduit margin. In contrast, we interpret the central shear zone as a shallow,
late-stage dilational structure, which partially tore the spine core with slight displacement. We explore
constraints from monitored seismicity and stick-slip behaviour to evaluate the rheological controls,
which accompanied the upward shift from compactional toward dilational shear as magma approached
the surface, and discuss their importance in controlling the permeability development of magma
evolving from overall ductile to increasingly brittle behaviour during ascent and eruption.

## 1.    Introduction

### 1.1.    Outgassing pathways and volcanic eruptions

The style and timing of activity exhibited during a volcanic eruption are strongly influenced by
the presence and mobility of volatiles in magma (Sparks, 1997; Woods and Koyaguchi, 1994) and
surrounding conduit wallrock (Jaupart and Allègre, 1991). During magma ascent, volatiles are exsolved
into gas bubbles (Navon et al., 1998; Sparks, 2003) as their solubility decreases with decompression
(Liu et al., 2005), crystallisation (Tait et al., 1989), and heat generated by crystallisation (Blundy et al.,
2006) and shear (Lavallée et al., 2015). This causes the accumulation of pressurised fluids in vesicles
that charges ascending magma, which, if sufficient may lead to fragmentation (Mueller et al., 2008;
Alidibirov and Dingwell, 1996) and an explosive eruption (Sahagian, 1999). The development of a
permeable network governs outgassing (Edmonds et al., 2003), pore pressure release (Mueller et al.,
2005), and eruptive cyclicity (Michaut et al., 2013), thereby reducing the potential for explosive activity
(Klug and Cashman, 1996) and encouraging effusion (Edmonds and Herd, 2007; Eichelberger et al.,
1986; Degruyter et al., 2012). Lava dome eruptions—the topic of this study—commonly switch
between effusive and explosive modes of activity due to this competition between permeability, pore
fluid pressure and the structural integrity of magma (Melnik and Sparks, 1999; Calder et al., 2015;
Cashman et al., 2000; Castro and Gardner, 2008; Edmonds et al., 2003; Lavallée et al., 2013; Lavallée
et al., 2012; Sparks, 1997; Holland et al., 2011; Kendrick et al., 2016; Platz et al., 2012). Considering
the water solubility-pressure relationships in magmas (Zhang, 1999), permeability-porosity
relationships in magma (Westrich and Eichelberger, 1994) and eruptive patterns (Edmonds et al., 2003),
it has been suggested that much of the outgassing during lava dome eruptions occurs in the upper few
kilometres of the conduit (Westrich and Eichelberger, 1994; Edmonds et al., 2003). This observation is
corroborated by rapid shallowing of seismicity leading to explosions (e.g., Rohnacher et al., 2021) and
the existence of shallow long-period seismic signals resulting from resonance in fractures and faults
(Chouet, 1996; Matoza and Chouet, 2010) as fluids are channelled to the surface (Holland et al., 2011;
Kendrick et al., 2016; Gaunt et al., 2014; Nakada et al., 1995; Newhall and Melson, 1983; Pallister et
al., 2013b; Sahetapy-Engel and Harris, 2009; Sparks, 1997; Sparks et al., 2000; Edmonds et al., 2003;
Varley and Taran, 2003; Stix et al., 2003). Therefore understanding the evolution of the permeable
network during eruptive shearing is central to constrain the evolution of the magmatic system in the
shallow crust (Blower, 2001).



Close examination of the architecture of shallow dissected conduits and structures in vent-
proximal silicic lava exposes complex shearing histories that would impact the permeable porous
network of erupting magma. These structures reveal porosity contrasts through the lavas, and strain
localisation near the conduit margins is commonly identified via the presence of flow bands and variably
porous shear zones with a spectrum of configurations (Gaunt et al., 2014; Kendrick et al., 2012;
Kennedy and Russell, 2012; Pallister et al., 2013a; Smith et al., 2001; Stasiuk et al., 1996; Tuffen and
Dingwell, 2005); features that are preserved to differing extents in crystal-poor and crystal-rich magmas
(Calder et al., 2015; Lavallée and Kendrick, 2021). For example, crystal-poor obsidian in dissected
conduits and dykes commonly exhibits marginal flow bands, showing alternation between glassy, finely
crystalline and microporous bands (Gonnermann and Manga, 2007). Flow bands also occur as variably
sintered, cataclastic breccia layers, resulting from fracture and healing cycles (Tuffen and Dingwell,
2005; Tuffen et al., 2003), and as variably sintered tuffisite layers, resulting from fragmentation and
entrapment of fragments into narrow fractures (Castro et al., 2012; Heiken et al., 1988; Kendrick et al.,
2016; Kolzenburg et al., 2012). Exposed crystal-poor conduits, dykes and domes are commonly dense,
as the porous network may easily collapse (unlike crystal-rich lavas; e.g., Ashwell et al., 2015). The
collapse of the porous network occurs as eruptions wane and pore pressure is insufficient to counteract
surface tension and local magmastatic and lithostatic stresses (Wadsworth et al., 2016a; Kennedy et al.,
2016), a process which hinders interpretation of the syn-eruptive permeable structure of crystal-poor
magma from the study of large-scale relict formations. Studies of erupted crystal-poor pumices (which
quench rapidly) help provide constraints on the extent of magma permeability at the point of
fragmentation (Wright et al., 2006), but the task of reconstructing the permeable architecture of an entire
conduit from these pyroclasts is challenging (Dingwell et al., 2016), further complicated by post-
fragmentation vesiculation (Browning et al., 2020) and vesicle relaxation (Rust and Manga, 2002), and
so remains to be attempted systematically.

Crystal-rich volcanic rocks (the subject of this study) expose a wider range of permeable porous
structures (Farquharson et al., 2015; Mueller et al., 2005; Klug and Cashman, 1996; Lamur et al., 2017;
Kushnir et al., 2016). For instance, dacitic volcanic spines extruded in 2004-08 at Mount St. Helens
(USA) and in 1994-95 Unzen volcano (Japan) reveal the presence of a complex 'damage halo' near the
conduit margin (Calder et al., 2015; Gaunt et al., 2014; Pallister et al., 2013a; Smith et al., 2001;
Kendrick et al., 2012; Wallace et al., 2019). These structures frequently share common characteristics,
with magma being increasingly sheared and damaged near the conduit margin, defined by a cataclastic
fault zone, adjacent to a brecciated wall-rock. A permeability study of the shear zone at Mount St.
Helens showed increased porosity and permeability, and the development of permeability anisotropy
towards the conduit margin (Gaunt et al., 2014), thus describing a scenario where shearing of dense,
crystal-rich magma induced dilation. In the case of Mount St. Helens' Spine 7, the fault zone is further
defined by the presence of a pseudotachylyte (Kendrick et al., 2012), a feature which can decrease the
permeability of fault zones (Kendrick et al., 2014a). At Unzen volcano, Smith et al. (2001) qualitatively
described the character of the shear zone developed in the centre of the lava spine at Mount Unzen,
highlighting the presence of a dilational cavity associated with shearing in the core of the magmatic
column. However, they did not quantify any porosity-permeability relationships. The cavity (hereafter
termed "central shear zone") was defined by an area in which the groundmass was torn, producing pore
spaces in the shadow of phenocrysts. The margin of the Unzen spine also hosts a spectrum of shear
textures (Hornby et al., 2015; Wallace et al., 2019), and significant low-frequency seismicity during the
eruption indicated flushing of fluids in the marginal fault zone (Lamb et al., 2015). Thus, the study of
evolving monitored signals and eruptive products at Unzen depicts a wide range of outgassing
pathways, which evolve during the course of magma ascent and lava dome eruptions.



*1.2. The permeability of magmas and rocks*


Several studies have explored the permeability evolution of volcanic materials, but due to the
occurrence of many influential structural and petrological processes in shallow volcanic conduits, no
solutions yet encompass the complete history of magma permeability during volcanic eruptions:
especially its time- and strain-dependent evolution. Following nucleation and growth, bubbles interact
and coalesce beyond a certain vesicularity, termed the percolation threshold, promoting the onset of
fluid flow through a connected bubble network (Baker et al., 2012; Eichelberger et al., 1986; Rust and
Cashman, 2004; Burgisser et al., 2017). The porosity of the percolation threshold varies widely
(between ~30 vol. % and 78 vol. % bubbles) depending on the size and geometry distributions of the
bubble population (Colombier et al., 2017; Rust and Cashman, 2004; Burgisser et al., 2017).
Vesiculation experiments have shown that permeability remains low in isotropically vesiculated
(aphyric and crystal-bearing) magmas as percolation initiates at vesicularities higher than those
theoretically predicted (Okumura et al., 2012; Okumura et al., 2009). Yet, bubble coalescence may be
accentuated by transport processes such as the thinning or draining of melt along the bubble wall (Castro
et al., 2012), deformation (Ashwell et al., 2015; Kennedy et al., 2016; Okumura et al., 2010; Okumura
et al., 2006; Okumura et al., 2008; Wadsworth et al., 2017; Shields et al., 2014; Farquharson et al.,
2016b; Kendrick et al., 2013), and rupture (Lamur et al., 2017; Lavallée et al., 2013; Heap and Kennedy,
2016; Okumura and Sasaki, 2014; Heap et al., 2015a; Laumonier et al., 2011), or lessened by fracture
infill (Kendrick et al., 2014a; Kendrick et al., 2016; Wadsworth et al., 2016b), all of which influence
the permeability of magma and promote permeability anisotropy (Farquharson et al., 2016c) during its
prolonged ascent to the Earth's surface.
In recent decades, laboratory measurements have helped us gain a first order constraint on the
permeability-porosity relationships of volcanic products (Eggertsson et al., 2018; Mueller et al., 2005;
Acocella, 2010; Rust and Cashman, 2011; Colombier et al., 2017; Farquharson et al., 2015; Klug and
Cashman, 1996). These suggest a non-linear increase of permeability with porosity; yet, depending on
the nature of the porous network, influenced by eruptive history, the permeability of rocks with a given
porosity may range by up to 4-5 orders of magnitude. Controlled laboratory experiments have given us
insights on probable permeability trends of magma subjected to different stress, strain, and temperature
conditions (Ashwell et al., 2015; Kendrick et al., 2013; Lavallée et al., 2013; Okumura et al., 2012;
Okumura et al., 2006; Shields et al., 2014), but a complete description of the dynamic permeability of
deforming magma requires *in-operando* determination under controlled conditions, which remain
scarce (Gaunt et al., 2016; Kushnir et al., 2017; Wadsworth et al., 2017; Wadsworth et al., 2021); these
studies have shown that surface tension and/or low-strain rate conditions under positive effective
pressure (*i.e.*, confining pressure greater than pore pressure) promote compaction and reduce
permeability. These informative descriptions require further inputs to enable robust relationships with
magma rheology, influenced by the presence and configuration of bubbles. Shallow magmas contain
bubbles and crystals and exhibit a non-Newtonian rheology (Caricchi et al., 2007; Lavallée et al., 2007;
Lejeune et al., 1999; Lejeune and Richet, 1995; Kendrick et al., 2013; Coats et al., 2018) that favours
the development of strain localisation, in particular, by preferentially deforming pore space (Kendrick
et al., 2013; Okumura et al., 2010; Shields et al., 2014; Pistone et al., 2012; Mader et al., 2013) As
magma shears, the porous network adopts a new configuration reflecting the stress conditions and
magma viscosity (Rust et al., 2003; Wright and Weinberg, 2009), which influences the permeability
(Ashwell et al., 2015; Kendrick et al., 2013; Okumura et al., 2010; Okumura et al., 2009; Okumura et
al., 2006; Okumura et al., 2008; Okumura et al., 2013). Shearing may increase or decrease the porosity
and permeability depending on the applied stress, strain and porosity of the deforming material and
direction of the permeability measurement due to the development of anisotropy (Ashwell et al., 2015;
Kendrick et al., 2013). In cases of extreme shear, magma may rupture, thereby increasing pore



connectivity and permeability (Laumonier et al., 2011; Lavallée et al., 2013; Okumura et al., 2013) until
the fracture heals via diffusion (Okumura and Sasaki, 2014; Tuffen et al., 2003; Lamur et al., 2019;
Yoshimura and Nakamura, 2010), seals via secondary mineralisation (Heap et al., 2019; Ball et al.,
2015), or infills with tuffisitic material (Castro et al., 2012; Kendrick et al., 2016; Kolzenburg et al.,
2012; Tuffen and Dingwell, 2005), which may densify through time (Kendrick et al., 2016; Vasseur et
al., 2013; Wadsworth et al., 2014; Farquharson et al., 2017). The densification of magma under isotropic
stresses (due to surface tension) has been reconstructed using high-resolution x-ray computed
tomography from synchrotron imaging, providing us with a first complete description of magma
permeability evolution as a function of porosity. This indicates that densification intrinsically relates to
the evolution of the size distribution and surface area of the connected pore space (Wadsworth et al.,
2017; Wadsworth et al., 2021). Nonetheless, a time- and strain-dependent description of the
development of the porous network of shearing magma remains incomplete, and information must be
sourced from our understanding of permeability evolution in deforming rocks.
In rock physics, the evolution of the porous network in deforming rocks has been extensively
studied. In its simplest description, the modes of deformation differ at low and high effective pressures
as rocks adopt brittle or ductile behaviour, respectively. These are defined as a macroscopic behaviour
(not a mechanistic description), whereby 'brittle' refers to the localisation of deformation leading to
rupture, and 'ductile' refers to the inability for rocks to localise strain during deformation (e.g., Rutter,
1986); see Lavallée and Kendrick (2020) and Heap and Violay (2021) for reviews of brittle and ductile
deformation in volcanic materials. The key distinction between these two deformation modes is that
brittle failure results in dilation (i.e. the creation of porosity), whereas ductile deformation results in
compaction of the porous network (Heap et al., 2015a). As a result, brittle (dilational) failure generally
enhances the permeability of rocks (Heap and Kennedy, 2016; Lamur et al., 2017; Farquharson et al.,
2016b), whereas ductile (compactional) deformation generally causes reduction in permeability (Heap
et al., 2015a; Loaiza et al., 2012). Despite its crucial role in defining deformation mode in rock, the role
of effective pressure in dictating the ductile and brittle modes of deformation has not been
systematically mapped out for multiphase magma; instead, we generally consider the effects of
temperature and applied stress or strain rate (e.g., Lavallée et al., 2008) over that of stress distribution,
as the deformability of magma imparts technical challenges to classic rock mechanic tests and
permeability determination (Kushnir et al., 2017). We may thus anticipate some similarities between
rock and magma deformation modes, whereby: At high effective pressure, ductile deformation is
favoured via compactant viscous flow or even cataclastic flow (if strain rates is relatively high to cause
pervasive fracturing of bubble walls), causing porosity and permeability decrease; at low effective
pressure, viscous flow may promote compaction at low strain rates whereas dilation may ensue if strain
rate favours localised rupture (Lavallée and Kendrick, 2020). Across this transition, magma rupture
may be partial and end abruptly, leaving a blunt fracture tip (Hornby et al., 2019). Most, if not all, of
the features observed in experimentally deformed rocks and lavas should be observable in a shallow
magmatic system hinging on a delicate balance between ductile and brittle deformation regimes,
promoted by outgassing which induces temporal and spatial variations in effective pressure. In this
study, we examine the well-preserved, dacitic lava spine erupted in 1994-95 at Unzen volcano to
constrain the permeability of dilational and compactional shear zones that developed in the shallow
volcanic conduit.

*1.3. 1990-1995 eruption of Unzen volcano*
Unzen volcano is a stratovolcano located near the city of Shimabara on the island of Kyushu,





Japan (Fig. 1). The volcano underwent a 5-year period of protracted dome growth which threatened the
surrounding population with the occurrence of several thousand rockfalls and many pyroclastic flows,
such as the destructive event on 3[rd] June 1991 that caused 43 fatalities. Activity initiated in early 1990
with a series of phreatic explosions and brief extrusion of a spine on 19[th] May; this was swiftly followed
by continuous growth of a lava dome until early 1995 (Nakada et al., 1995). Between October 1994 and
January 1995, the eruption concluded with the extrusion of a spine through the dome surface (Fig. 1c).
At the dome surface, gas emissions focused along the spine marginal faults (Ohba et al., 2008). The
dome products have a dacitic composition and contain euhedral phenocrysts of plagioclase and
amphibole in a groundmass containing microlites of plagioclase, amphibole, pyroxene and iron oxides
(Nakada et al., 1995; Wallace et al., 2019). Petrological constraints suggest that degassing initiated at a
pressure of approximately 70-100 MPa; *i.e.*, in the upper ~3-4 km depth (Nakada et al., 1995).
Dome growth occurred in stages, forming thirteen discrete lobes until mid-July 1994. Growth
was observed to be typically exogenous when effusion rates were high, and endogenous at effusion
rates lower than 2.0 x 10$^5$ m$^3$d$^{-1}$ (Nakada et al., 1999). In five years, the eruption generated 2.1 x 10$^8$ m$^3$
of lava at an average ascent rate estimated at 13–20 md$^{-1}$ (Nakada et al., 1995); the final spine extruded
from late-1994 to early-1995 at a rate of approximately 0.8 md$^{-1}$ (Yamashina et al., 1999). The rheology
of the erupted dome lavas has been sourced of debate (Goto et al., 2020; Sato et al., 2021), as it is
challenging to precisely reconstruct the physico-chemical, petrological and structural parameters which
control rheology as a function of depth during eruption. For the late-stage spine, Nakada and Motomura
(1999) proposed that it formed due to a lower effusion rate, which resulted in extensive magma
degassing and crystallisation, and thus high viscosity, which promoted rupture and exogenic growth at
relatively low strain rates (e.g., Hale and Wadge, 2008; Goto, 1999). Extrusion occurred through
pulsatory magma ascent, accompanied by ~40 h inflation/deflation cycles (Yamashina et al., 1999) and
a rhythmic pattern of summit earthquakes, interpreted to result from magma rupture in the top 0.5
kilometre of the conduit (Lamb et al., 2015; Umakoshi et al., 2008); waveform correlation of the seismic
record revealed rhythmic seismicity grouped into two primary clusters (Lamb et al., 2015). Hornby et
al. (2015) statistically analysed the slip duration of seismic events in the clusters, defining a mode and
mean of 0.1 s. As magma ascent occurred through an inclined conduit (Umakoshi et al., 2008), the spine
extruded at an inclined angle of ~45º towards the ESE (Fig. 2a) and increasingly leaned against the
lower fault zone as extrusion rate waned, causing the shallowing of seismogenic magma rupture in this
area (Lamb et al., 2015). In contrast, the upper fault zones may have opened up as the spine settled, thus
triggering rupture at increasing depth and promoting preferential pathways for fluid flow (Lamb et al.,
2015). By the end of the eruption, the spine achieved approximate dimensions of 150 m length, 30 m
width and 60 m in height (Nakada and Motomura, 1999; Nakada et al., 1999); it is complemented by
multiple fragments of spines, extruded earlier in the eruptive phase, which we examine in this study.
Unfortunately, the lower and upper fault zones are not observable in the spine exposures, but the
northern lateral conduit margin contains well-defined shear zones (Smith, 2002; Smith et al., 2001),
which are revisited here and augmented by structural and microtextural descriptions as well as porosity
and permeability constraints. Our study of the spine sheds new light on the permeability evolution of
its shear zones, and thus the nature of outgassing during the waning phase of the 1990-1995 eruption.

**2.      Materials and Methods**
*2.1      Localities and sample collection*



The 1994-95 lava spine was investigated during two field campaigns, in November 2013 and
May 2016. Close structural examination at different scales forms the basis of this study along with
porosity and permeability measurements, using field and laboratory equipment. Owing to the inclination
of the spine (extruded towards the east), large blocks ranging from 5 to 20 m-wide are dislocated from
the front of the *in situ* western main spine structure (Fig. 2a, b). Here, we investigated two blocks that
reveal a central shear zone (CSZ) and marginal shear zone (MSZ) that developed in the spine. These
detached, yet fully intact, spine blocks were selected owing to their contrasting shear textures that would
have represented different positions within the volcanic conduit during magma ascent and extrusion
(i.e., central vs. marginal), thus allowing assessment of syn-eruptive outgassing pathways. The marginal
shear zone (MSZ) block, located ~60 m east of the main spine (Latitude: 32.76131º Longitude:
130.29983º), was carefully sampled to quantify the spatial distribution of permeability across the spine
margin (samples A-H; Fig. 2c). The central shear zone (CSZ) block, located centrally between the main
spine and MSZ (Latitude: 32.761271º Longitude: 130.299472), features the dilatational cavity
(described in Smith et al., 2001) and was also studied *in situ*, using non-destructive methods to preserve
the integrity of this exemplary feature. The main spine and CSZ are protected by UNESCO heritage
site regulations (Figs. 1c, 2a), thus only permitting *in situ* sample collection from the MSZ.

*2.2    Sample preparation*
Samples collected from the marginal shear zone were cut and cored parallel to the shear
direction and perpendicular to the shear plane in order to constrain the anisotropy developed in shear
zones. A total of eight thin sections (fluorescent dyed) were prepared for microtextural analysis (labelled
A-H). For the largest samples (A, B, C, E, H; see Fig. 2c-d) a set of 2-3 cylindrical cores (two parallel
and one perpendicular to shear plane) were prepared with a diameter of 26 mm and a length of 30 or 13
mm, depending on the size of the sample. Within the highly sheared sample B (Fig. 2c-d), which is
directly adjacent to the fault and gouge zone, multiple sets of cores of 20 mm diameter were prepared,
closely spaced, to obtain porosity/permeability determinations at a higher resolution across this defining
part of the shear zone.
*2.3    Microstructural analysis in 2D and 3D*
2D analysis of the microstructures exhibited across the shear zones was carried out using a
Leica DM2500P optical microscope in plane polarised and ultraviolet (UV) light, as well as a Philips
XL30 scanning electron microscope (SEM) in backscattered electron (BSE) mode, set at 20 kV and 10
mm working distance. For this purpose, representative features were imaged for each sample across the
shear zone (Fig. 3).
To further evaluate the architecture of the porous network in three dimensions (3D), four
samples collected across the shear zone were scanned using a phoenix nanoton® m x-ray computed
tomography scanner to produce high-resolution reconstructions with a voxel size of 11.111 μm. For
each sample we acquired 1440 radiographs, scanning 360°, under the following conditions: exposure
time of 1000 ms; voltage of 80 kV; current of 120 μA; 0.2 mm aluminium filter. The radiographs were
then reconstructed using the inverse Radon transformation (Radon, 1986), resulting in a 3D image of
the sample. These files were processed in FEI Avizo and ImageJ/Fiji software to illuminate the
permeable, porous network.

*2.4.    Porosity measurement in the laboratory*




Each core was dried in an oven at 50 °C overnight, then kept in a desiccator (for thermal
equilibration to ambient conditions) before being weighed and loaded in a pycnometer. The fraction of
connected pores (which controls permeability; Colombier et al., 2017) was determined using a
Micromeritics AccuPyc II 1340 helium pycnometer. The porosity determination first necessitated
measurement of the geometric volume of the sample ($V_{sample}$). Then, once inserted in the specimen
chamber of the pycnometer, helium gas was injected in the chamber to estimate the volume taken up
by the solid fraction of the sample, thus providing the skeletal volume ($V_{skeletal}$) of the rock. The
fraction of connected pores ($\phi_{connected}$) in a sample was then calculated via:
$$\phi_{connected} = \frac{(V_{sample} - V_{skeletal})}{V_{sample}}$$    (1).

*2.5.    Permeability determination in the laboratory*
The prepared cores were jacketed with a Viton™ tube and inserted in a hydrostatic cell from
Sanchez technologies to measure permeability and pore volume as a function of pressure. The jacketed
samples were externally loaded using a Maximator® oil pump to various confining pressures ($P_c$) and
internally loaded using water to an average pore pressure ($P_p$) of 1.25 MPa, in order to obtain a range
of effective pressures ($P_{eff} = P_c - P_p$) from 5 to 100 MPa. Each time the sample was loaded to new
confining pressure increment, the volume of water expelled from the pores in a given sample (due to
compaction) was monitored to constrain pore volume change due to crack closure as a function of
pressure (Lamur et al., 2017). Steady-state flow permeability *(k)* was measured by applying low pore
pressure gradients *(ΔP)* of 0.5 and 1.5 MPa to ensure laminar flow with no slip conditions (Heap et al.,
2017a) to satisfy Darcy's Law:
$$k = \frac{Q\eta L}{A(\Delta P)}$$    (2),
where $Q$ is the flow rate monitored through the sample (m³s⁻¹), $\eta$ is the viscosity of the water in pores
(Pas), $L$ is the length of the sample (m), and $A$ is the cross-sectional area of the sample (m²).

*2.6    In-situ permeability measurements in the field*
To measure the permeability of rocks in the central shear zone (CSZ; Fig. 1c) that could not be
sampled for laboratory testing due to preservation restrictions, we used a non-destructive, portable, air
permeameter (TinyPerm II) from New England Research, which estimates permeability by monitoring
pressure recovery rate from a vacuum, based on the concept of transient pulse permeability (Brace et
al., 1968). The apparatus is hand-held and needs to be employed carefully to maintain a consistent seal
between the nozzle of the permeameter and rock surface throughout the measurements (lasting up to a
few tens of minutes). It may be used to determine the permeability of rocks between approximately 10⁻
¹² to 10⁻¹⁶ m² (Farquharson et al., 2015; Kendrick et al., 2016; Lamur et al., 2017). In this study, three
transects were measured across the central shear zone and all measurements were performed twice to
ensure precision of the method (as determined in Lamur et al., 2017).

**3.    Observations and results**



The 1994-95 spine structure at Mount Unzen is exposed in several large segmented blocks (Fig.
1c-d; Fig. 2a-b). A thorough structural description of the main spine structure and subsidiary block (e.g.,
CSZ) can be found in Smith et al. (2001); here we highlight the main features. The lava spine is split
into a few very large, primary blocks, ~20-30 m wide and high (Fig. 1c-d, 2a-b), broken roughly
perpendicular to extrusion direction: westward and inclined (see Fig. 2b). The CSZ block seen in Figure
1c shows a >8-m wide variably deformed core (I) lying adjacent to a 2-m wide intensely sheared zone
(II), bordered to the north by a dextral fault and coupled to a large indurated breccia (III), uplifted from
the surrounding dome emplaced. The lower and southern edges were not exposed. The upper edge of
the spine was not accessible, but we noted large, incoherent brecciated blocks. The rear of this outcrop
as well as the main *in situ* spine structure exhibit irregular, metre-scale polygonal joints, although these
are not developed in the face of the outcrop studied here (Fig. 1c). Additional fragments of the spine
occur in a few subsidiary blocks (e.g. Fig. 1d), located a few tens of meters to the east of the main spine
(Fig. 2a). These blocks, which were emplaced prior to the main spine, expose several sections through
the spine, and reveal the evolving architecture of the shear zone in the shallow magmatic conduit. One
such block, shown in Figure 1d, exhibits a ~1-m wide shear zone, bordered to the left by a set of oblique
tensile fractures, reaching 2-5 m in length and spaced at ~ 3 to ~10 cm intervals, and to the right by an
indurated breccia. This prominent block was not sampled or further studied to preserve its integrity.

### 3.1    *The marginal shear zone*

### 3.1.1.    *Structural and microtextural observations*

Our primary field location for this study was a 4.7-m wide block of the spine, exposing the
northern marginal shear zone consisting of gouge, sheared lava and the spine core (Fig. 2c-d). The
outcrop displayed mild surface weathering, in the form of a thin (micron-size) veneer of unknown
precipitate on the rock surface (which was inclined at an angle of ca. 40° towards the West). This thin
veneer did not visually obstruct any primary magmatic textures and structures, and the shear texture
was clearly visible, yet we it would prevent accurate field permeability constraints. Four distinct degrees
of shear were visually defined through textural examination and changes in surface roughness across
this section of the conduit (Fig. 2c-d): a fault gouge zone (sample A) bordering a high-shear zone
(samples B, C, D), a moderate-shear zone (samples E, F) and low-shear spine core (samples G, H) in
decreasing order of surface roughness and visually observable fracture density variations; quantitation
of fracture density was not attempted as we deemed the thin veneer may have prevented meaningful
accuracy. This shear-based division is consistent with a complementary investigation of the
mineralogical characteristics of this shear zone (Wallace et al., 2019). The contacts between shear zones
trend approximately E-W in the outcrop (Fig. 2c,d), and so roughly parallel to the spine emplacement
direction to the ENE, despite the detachment of this spine block from the main intact spine body to the
west. Eight samples were systematically collected across this shear zone for further analysis (labelled
A-G in Fig. 2c,d): eight for 2D microstructural analysis (PPL, UV light and BSE imagery; Fig. 3), four
for tomographic imaging (Fig. 4) and five for porosity and permeability determination (Fig. 5-6). [Note
that multiple cores were obtained from the five blocks sampled for laboratory measurements.]
The spine core, termed low shear herein (~1.5 m wide; Fig. 2c, d), exhibited a smooth surface
and the phenocrysts showed no preferred orientation at the macroscopic scale. In samples G and H
collected from the low shear zone (Fig. 3), phenocrysts of plagioclase, amphibole, biotite (plus minor
quartz) are typically euhedral, largely intact and up to ~5mm in length (Fig. 3); groundmass microlites
also show no preferred orientation in BSE images. The porous structure is characterised by a diktytaxitic
texture, composed of some large, irregular, vesicles with 'ragged' edges, appearing intrinsically related



to the presence of surrounding phenocrysts (single white arrows on UV light images in Fig. 3). Small
fractures are often seen to originate from these large vesicles, penetrating pervasively through both
phenocrysts and the groundmass (double white arrows in Fig. 3). The groundmass contains abundant
small vesicles, showing a high degree of connectivity as revealed by tomography (Fig. 4g-h).

The moderate-shear zone is approximately 2 m wide (Fig. 2c, d). In this zone, we observed an
increased fracturing of phenocrysts and changes in the distribution of porosity. Scrutinising the sample
E under microscopy, we observe that the phenocrysts, which rarely exceed 2 mm in size in this zone,
are commonly micro-fractured (Fig. 3). The vesicles are occasionally large and connected (Fig 3, 4e-f),
and while the vesicular texture remains diktytaxitic (as in the low shear spine core), the vesicles in
sample E appears increasingly aligned and localised around phenocrysts as the magnitude of shear
increases towards the fault; similarly, the microlites show increasing degrees of alignment (revealed by
undulose extinction angles; see Wallace et al., 2019). Thin bands (<200 μm width) of reduced porosity
are observed to localise in the groundmass (see facing double arrows in UV light images in Fig. 3),
which are notably absent in the low shear zone; these are (sub-)parallel to the shear plane. The
tomographic reconstructions show irregular vesicles, which are surrounded by fractures and invaded by
rock fragments (Fig. 4e). These vesicles enhance the connectivity of the porous network (Fig. 4f).

The high shear zone is approximately 1 m wide (Fig. 2c, d) and marks the beginning of micro-
and meso-scopic shear bands, at a scale of the order of a few millimetres, near-parallel with the direction
of shear; these increase in abundance and scale nearer the fault, especially within the final 0.1-0.2 m
(see features denoted in Fig. 2c-d as well as enlarged in the inset). The bands, which form a pervasive
foliation (S), consist of elongate, white porphyritic plagioclase lenses, fractured and crenulated. The C-
S fabrics are parallel in this area. These porphyritic bands are flanked by reddish-brown groundmass as
well as thin, elongate biotite phenocrysts (see sample B "fresh surface" in Fig. 3). The plagioclase and
biotite commonly exhibit a mineral fish texture. Under the microscope, we observe that the biotite show
undulose extinction from crystal plastic deformation (see Wallace et al., 2019, for a detailed crystal
plasticity study). Intense banding (observed as faint lineations of reduced porosity under UV light in
the moderate shear zone; Fig. 3) is observed adjacent to, and running parallel with, the fault-gouge
contact. The bands are up to 1 mm wide and display variations in porosity under UV light (Fig. 3), as
also revealed by tomography (Fig. 4c-d). The dense bands are traversed by hairline fractures a few
hundred microns in length and contain a few isolated millimetre-size vesicles, generally adjacent to
large phenocryst fragments (samples B and C in Fig. 3). More porous bands display disordered and
fragmental textures (sample B), with abundant, irregular large pores and cracks, and pulverised
phenocrysts (PPL and UV light in Fig. 3); macroscopically, the most porous bands often appear like
ragged tensile fractures. The transition between dense and porous bands is abrupt, occurring over a few
tens of microns (BSE images of samples B and C in Fig. 3). Microlites and microphenocrysts are aligned
with the banding, and thus with shear and extrusion direction (Fig. 3). The high shear region of the
spine is further crosscut by multiple sub-parallel curvilinear extensional bands (i.e., weakly defined
fractures), up to ~1 m in length, and trending ~57˚ from the primary C-S fabrics in a Riedel-like fashion
(Fig. 2c, d); some of these bands extend into the moderate shear zone but only faintly. These bands,
spaced by 3-6 cm (~4.5 cm in average), show opening of ca. 1-2 mm in places. [Note that the blue traces
in Figure 2 denote the general attitude, not the spacing, of the bands]. The Riedel fractures appear to be
associated with a set of faint, conjugate fractures (R'), although their observation is not ubiquitous
across the high-shear zone.

The fault zone hosts up to ca. 0.2-m thick gouge material (Fig. 2c,d). The contact between the
gouge and the high shear zone is generally sharp, and often planar, although we observed small
embayments, especially along C-S fabrics in the neighbouring high shear zone (Fig. 2d). [Note that the



extent of the gouge is not exposed equally across the outcrop as material was likely lost during
separation of this block from the main spine upon eruption; so the surface does not reflect the contact
geometry. This material loss also led to obliteration of vestiges of a pseudotachylyte, suggested by local
partial melting textures presented by Wallace et al. (2019)]. The gouge is typified by well-consolidated,
fine-grained cataclasite with some larger rounded clasts up to ~15 mm in diameter (sample A; Fig. 2c
inset). The gouge is matrix supported and displays a strong foliation parallel to spine extrusion direction.
Conjugate fractures form a dominant feature contributing to the porosity of the gouge. Microscopically,
the rock is pervasively fragmented (sample A in Fig. 3); the few phenocrysts that remain relatively
intact often display signs of deformation. The fragments in the gouge are generally densely compacted
and the porosity is uniformly distributed, with little banding or preferred orientation of fragments at the
microscopic scale, although connected pores occasionally exhibit a degree of alignment at small scale
(Fig. 3) and at large scale as observed via x-ray tomography (Fig. 4a-b).

### 449 3.1.2 Connected porosity across the marginal shear zone

The porosity of the rocks, determined via pycnometry, indicates variations between 8 % and 27
% across the shear zone and in the fault gouge; Figure 5a displays the average of multiple measurements
from the different cores prepared from each sample. The measurements indicate that the high shear zone
generally holds slightly lower porosities than surrounding areas. Within the high-shear zone (sample B)
we measured important variations in porosity ranging between 8 % and 15 % due to flow bands (e.g.,
in sample B); yet, the coarseness of sample measured prevent from accurately quantifying the highly
variable degrees of porosity visually observable in hand specimen.
When loading the samples (cored parallel with to spine extrusion direction) in the hydrostatic
pressure vessel, we observed a nonlinear decrease in porosity of up to 4 % by increasing the effective
pressure to 100 MPa (Fig. 5b). The data shows a similar dependence of porosity on effective pressure
for the coherent samples from the low, moderate and (densest part of) high shear areas, with a slightly
larger reduction in porosity with effective pressure in the initially most porous, high shear bands and
granular gouge sample (Fig. 5b).

### 464 3.1.3 Permeability across the marginal shear zone

The permeability of the rocks collected across the spine segment reveals a ~1-m wide region of
low permeability in the high shear zone, compared with the moderate shear zone, the low shear spine
core and fault gouge (Figs. 5, 6). There appear to be abrupt variations in permeability (decrease and
increase) in sheared rocks directly adjacent to the fault gouge, due to the alternation between dense and
porous shear bands.
The data show considerable differences in the permeability parallel and perpendicular to the
plane of shear (Fig. 3c,d) across the shear zone (Fig 6a,b). In the high shear zone permeability was
found to be higher in the plane of shear (i.e., parallel with extrusion direction) than perpendicular to it,
whereas in the moderate and low shear zones, as well as in the gouge, permeability was essentially
isotropic. Anisotropy is cast here as a ratio between the permeability parallel and perpendicular to the
shear plane (Fig. 6c). The anisotropy is most pronounced in the high shear zones, where, in one instance,
the permeability ratio increases dramatically from three to over seven times larger parallel than
perpendicular to the shear plane with increasing confining pressure in a hydrostatic pressure vessel (Fig.
6c). In other samples, the anisotropy increase with pressure is less or even negligible, indicating the



heterogenous nature of the high shear zone. This sensitivity to confinement is due to the presence of the
distinct dense and porous bands in the sheared lava (Fig. 5b, 6); in the cores parallel to the shear plane,
fluid can flow through porous bands from top to bottom of the sample, whereas perpendicular to shear,
fluids must pass through both and dense and porous bands to traverse the sample. Fluid flow in the
denser areas will be dominated by channelling through narrow fractures (sub-horizontal in BSE images
in samples B and C in Fig. 3), which are more susceptible to closure by increasing effective pressure
than equant pores (e.g., Kendrick et al., 2021). Although this process occurs during confinement in both
orientations, it only impacts permeability perpendicular to shear direction, and so contributes to
enhanced anisotropy of permeability in banded shear fabrics under confinement (Kendrick et al., 2021).
*3.2   Central shear zone*
*3.2.1   Structural observations*
The second feature of interest is the cavity exposed in the central shear zone block (Fig. 1c and
2a). This section of the spine has been described in detail by Smith et al. (2001); here, we review key
aspects observed in the field as no samples were collected to conserve the exposure of this world-class
feature. We only examined the rocks forming this structure and performed non-destructive, *in-situ*
testing.
The central shear zone (CSZ) is located near the centre of the spine core (Fig. 1c). Its primary
feature is the presence of a porous cavity, which curves and pinches out (upward) from the end of a
dominant, 9-cm wide fracture, extending approximately 3 m in length (determined from the visible
extent of the exposure). Unlike the aforementioned marginal shear zone, which displays an increased
degree of shear towards the spine margin, the central shear zone exhibits an increase in shear towards
the centre of the spine. From left to right (i.e., northward) on Figure 7, we note an increase in aligned,
bent and broken phenocrysts as well as aligned shear bands (ostensibly parallel with the dominant
fracture), fractures and surface roughness, which terminates upon intersecting the end cavity; beyond
which point, the rocks show no clear evidence of shear, including shear bands, elongate pores or aligned
crystals. This is evident in the field photograph (Fig. 7) as steeply inclined porous bands which ends
against the southern (i.e., right) side of the cavity;  on the southern side the sheared lava exhibit a higher
porosity than the surrounding undeformed rocks (although this could not be quantified in the field).
Approximately 1 m above the pinched-out tip of the main cavity, we observe the presence of a
secondary porous cavity (Fig. 1c inset), approximately 60 cm long, and elongated parallel to the fracture
that connects to the main cavity.

*3.2.2   Permeability across the central shear zone*
The permeability of the rocks in the central shear zone was measured along three transects in
two field campaigns (in November 2013 and May 2016) to negate potential influence from variable
degrees of water saturation of the rocks at different times of year. Our field measurements are consistent
with one another. The permeability varies very little in the undeformed areas of the outcrop (i.e., on the
right-hand side of the fracture in Fig. 7) for all transects, with an abrupt increase in permeability up to
three orders of magnitude in the 9cm wide central cavity, and elevated permeability in the ~40 cm wide
proximal sheared area to the left of the fracture.

**4.    Interpretation**



The contrasting permeability, porosity and (micro)structural changes observed across the
marginal and central shear zones reveal the impact of shear and distinct modes of magma deformation
during shallow conduit ascent. Here we interpret each of these key features for the development of
volcanism at lava domes.
*Marginal shear zone*

The marginal shear zone is characterised by a 3-m wide zone in which strain caused changes in
the porous structure, via crushing of the pore walls as well as distortion and failure of the crystalline
phase; these promoted an increased reduction in pore volume and permeability towards the fault,
especially in the high shear zone. Smith et al. (2001) invoked the effects of gravitational forces during
post-emplacement flow of the lobes as a mechanism for the development of 'ragged' pores and
porous/dense flow banding in dome lavas at Unzen volcano. Yet, such diktytaxitic structure have been
observed in small surficial dome blocks at Santiaguito volcano (Guatemala), which have not suffered
from gravitational effects associated with flow along the flanks (Rhodes et al., 2018). This diktytaxitic
texture has been observed in the experimental products of lavas compacted under uniaxial (Ashwell et
al., 2015) and triaxial (Kushnir et al., 2017) conditions. Similarly, they can be reproduced (to a high
degree of similarity) through shear-enhanced compaction of porous rocks under high effective pressures
(Heap et al., 2015a; Heap et al., 2015b). The commonality between these experiments is that they were
carried out in the ductile field, through which material may sustain substantial compaction without the
propensity for developing localised strain (Rutter, 1986) – a regime that results in a permeability
reduction through shear (Ashwell et al., 2015; Kushnir et al., 2017; Heap et al., 2015a; Heap et al.,
2015b). In this regime, magma deformation may result in crystal plastic distortion and failure (Kendrick
et al., 2016), as witnessed at Unzen (Wallace et al., 2019). Thus, we interpret the bulk of the marginal
shear zone as the result of ductile deformation, which resulted in distributed, pervasive shear over a
width of 3 m. Within this part of the conduit, the high shear zone displayed the highest degree of shear-
enhanced compaction.
However, ductility alone is insufficient to describe the marginal shear zone. For instance, the high-shear
area exhibits a foliation (S plane) and fractures (C plane) parallel to the shear plane, which is then
crosscut (parallel but undulating) by a marginal fault hosting gouge formed by comminution and
cataclasis, containing conjugate fractures. The composite C-S fabric in the high shear zone is
increasingly penetrative towards the fault core (at the gouge contact), and its parallel C and S planes
indicates that the shear zone accommodated significant strain. This is supported by observation that
curvilinear Riedel fractures have developed and overprinted the C-S fabric at an angle of 57˚ (cf.
Ramsay, 1980). Such an angle is consistent with a lava body undergoing rupture following sustained
ductile deformation (e.g., Lavallée et al., 2013); it is also consistent with the progressive thickening of
a shear zone formed via simple shear with a small component (<10 %) of pure shear (assuming pure
and simple shear are planar; Fossen and Cavalcante, 2017); this minor pure shear component is further
supported by the presence of weakly defined conjugate fractures crosscutting the Riedel fractures. Both
the gouge and the fractures through the high shear zone were constrained to have locally higher
permeability and porosity than the bulk of the shear zones: features characteristic of dilational
deformation resulting from macroscopically brittle failure (Heap et al., 2015a; Heap et al., 2015b;
Laumonier et al., 2011). Riedel fractures generated in experimentally deformed magma have been
described as important pathways to redistribute fluids across shear zones (Laumonier et al., 2011), and
we anticipate the impact would be similar at Unzen; the Riedel fractures in the marginal shear zone only
reached ~1m in length, but the marginal shear zone in other blocks (Fig. 1d) contain oblique Riedel
fractures that reach 2-5 m in length (Fig. 1d) which would have formed efficient fluid flow pathways.
Thus, we interpret the marginal shear zones to reflect the evolution of magma shearing across the ductile
to brittle transition during shallowing of the ascending spine, which impacted fluid flow during eruption.



*Central shear zone*

The central shear zone detailed in this study has a very different character. Macroscopic observations of numerous cracks suggest that it is dominantly dilational, as supported by the drastic increase in permeability towards the fault and cavity. Despite having opened by ~9 cm, the main fracture tip is blunted as it terminates in a curvilinear cavity, and seemingly disappears before reappearing as a secondary cavity 1 m above (Fig. 1c inset). This is akin to areas of reduced density that develop ahead of a crack tips during material failure in the lab (e.g., Célarié et al., 2003) and indicates immature shear that was insufficient to enable the continuous propagation of a fault across the whole spine. This, in conjuncture with the observation that shear becomes more pronounced towards the centre of the spine, suggests that the areas undergoing shear may have locally shifted towards the conduit core; yet, displacement was not extensive. The reason for this shift is difficult to assert, but we posit that the shallow calving of blocks from the spine front, progressive inward cooling and/ or the higher porosity of the magmas in the conduit core (compared to a denser, compacted and strained conduit margin) may have shifted the locus of deformation towards the conduit core at the end of the eruption.

The shear zones studied here indicate that the dominant deformation regime of magma may evolve spatially and temporally during ascent in volcanic conduits, which would modify the magma's permeability and its ability to localise and channel outgassing during the effusion of lava domes.

## 5.  Discussion

*Permeability in volcanic environments*

The power of volcanic eruption models relies on an understanding of the coupling between magma and volatiles in volcanic conduits (Sparks, 1997), yet a description of dynamic permeability of deforming magma eludes us. The studies of eruptive products have provided first order constraints on the relationship between permeability and porosity (Fig. 8; Klug and Cashman, 1996; Mueller et al., 2005; Farquharson et al., 2015) for various types of volcanic rocks (e.g. explosive clasts vs effusive lavas), including the presence of heterogeneous structures (Farquharson et al., 2016c; Kolzenburg et al., 2012; Lamur et al., 2017; Kendrick et al., 2021), and these constraints have been invoked in diverse models to assess how magma permeability may evolve leading to eruption (Burgisser et al., 2019; Edmonds et al., 2003). However, the deformability of magma imposes constant changes to the porous permeable network and to date, only a few studies have measured or assessed the transience of permeability and porosity during magma deformation (Okumura et al., 2010, 2012; Kendrick et al., 2013; Ashwell et al., 2015; Kennedy et al., 2016), especially *in operando* (Kushnir et al., 2017; Wadsworth et al., 2017; Wadsworth et al., 2021). Considering the range of pressure conditions (e.g., pore pressure gradient, local deviatoric stress) and magma properties, none of these studies has yet succeeded in fully reconstructing the evolution of porosity and permeability of magma shearing during ascent in volcanic conduits.

The rocks sampled across the shear zone and in the fault gouge at Mount Unzen vary in porosity between 8 % and 27 %; this range is slightly narrower than the porosity range (4-48 %) covered by blocks shed by pyroclastic density currents originating from the domes during the 5-year eruption (see Fig. 8; Kueppers et al., 2005; Coats et al., 2018; Kendrick et al., 2021; Scheu et al., 2007; Mueller et al., 2005). The narrower range exhibited by the spine shear zones may reflect the occurrence of fewer porosity-modifying mechanisms (e.g. post-fragmentation vesiculation) in the highly viscous spine lava compared to those which occurred throughout the entire course of the eruption, which are represented by the blocks at the foot of the volcano. We see the largest contrast when we compare the permeability



range of the lavas which erupted through the spine at the end of the eruption (~$10^{-15}$ to ~$10^{-14}$ m$^2$, at the
lowest effective pressure) with that obtained from rocks recovered by drilling through the eruptive
conduit at a depth of ~1.5 km (~$10^{-17}$ to ~$10^{-19}$ m$^2$) in the framework of the Unzen Scientific Drilling
Project, drill hole 4 (USDP-4) (Watanabe et al., 2008). The latter rocks, originating from magma stalling
at depth, reflect greater time under compactant conditions and porosity infill and reduction from
secondary mineral precipitation (Yilmaz et al., 2021). The large difference in permeability between the
two datasets alludes to the highly variable spatial and temporal variation of magma permeability within
even a single volcanic system.
Previous investigations of permeability in shallow volcanic conduits have highlighted the
existence of dilational shear zones, whereby the conduit margin is bound by a permeable 'damage halo';
this has been proposed through both field (Saubin et al., 2019; Pallister et al., 2013a; Gaunt et al., 2014;
Wallace et al., 2019) and laboratory (Lavallée et al., 2013; Laumonier et al., 2011) studies. These
constraints indicate a high-permeability zone, with a strong component of anisotropy, with fluid flow
preferentially developed in the direction of extrusion due to shear fabrics (Wright et al., 2006; Gaunt et
al., 2014; Wallace et al., 2019). Connectivity is enhanced by fractures, which would contribute to the
development of anisotropy and preferential channelling of fluids along the conduit margin, promoting
concentric or ring-like gas emissions, as for instance exemplified at Santiaguito, Guatemala (Lavallée
et al., 2013). Here, at the conduit centre at Unzen we observed a localised dilational shear zone up to
three orders of magnitude more permeable than the surrounding magma. This zone spans a relatively
narrow section of the conduit and appears to be a late, immature feature that is possibly related to shear
during the final stages of ascent and/ or structural readjustment during failure and calving of portions
of the spine to the ENE. Instead, the primary (and volumetrically most significant) marginal shear zone
studied at Unzen is mostly compactional and exhibits a lower permeability than the surrounding magma,
particularly in the plane perpendicular to shear direction. It appears to have formed at depth, before
being overprinted by shallower faulting. Seismic analysis indicated that seismogenic faulting was
episodic and shallow, likely originating in the upper 500 m of the conduit (Umakoshi et al., 2008; Lamb
et al., 2015); the pulsatory magma shearing above this depth would have resulted in switches between
compactional and dilatant shear, causing locally higher permeability fractures through the sheared
magma, and a permeable marginal fault gouge by cataclasis (Fig. 9). Such intermittent seismic stressing
may also serve to weaken surrounding country rocks and modify permeable pathways (Schaefer et al.,
642  2020).


*Ductile-brittle transition in ascending magma*

The presence and overprinting of compactional and dilational shearing modes in close
proximity in a given magmatic extrusion demands appraisal. The ductile-brittle transition of materials
has long been studied and is generally better understood for rocks than lavas as more low-temperature
tests have been carried out (Paterson and Wong, 2005; Rutter, 1986; Heap et al., 2015a). Reconstruction
of yield caps (or curves), based on the shear stress required for rupture or flow of materials at different
effective mean stress, have shown that porous rocks undergo a transition from macroscopically brittle
to ductile deformation modes with increasing effective pressure (Fig. 9b); this transition sets in at lower
effective pressure (i.e., either at shallower depths or with higher pore pressures) if the material is more
porous (Heap et al., 2015a; Coats et al., 2018). However, magma is viscoelastic, thus depending on the
timescale of observations magma may behave as a solid; in essence, as a rock. Magmas abide to the
glass transition so that at long observation timescales or under slow deformation, they flow; but at short
timescales or if strain rate is high, they may rupture (Dingwell, 1996). The strain rate to meet this



transition decreases if melt viscosity increases due to cooling, crystallisation, degassing, and/ or
vesiculation (Wadsworth et al., 2018; Dingwell and Webb, 1989, 1990; Cordonnier et al., 2012;
Cordonnier et al., 2009; Coats et al., 2018; Lavallée et al., 2013; Lavallée et al., 2008). The glass
transition of silicate melts, which controls the deformation mechanisms of magmas (viscous or brittle),
thus impacts their deformation modes, brittle or ductile (be it viscous flow or cataclastic flow);
applicability of the concept of yield caps to volcanic rocks and magma, as shown in Figure 9b, have
been reviewed by Lavallée and Kendrick (2020). In a scenario where magma ascends, deforms and
outgasses during an eruption, such as during spine extrusion at Unzen, magma may undergo a transition
from a macroscopically ductile to brittle deformation mode due to a reduction in effective pressure
(from ascent or due to pore pressure increase; Heap et al., 2017b), densification (Heap et al., 2015a;
Coats et al., 2018), viscosity increase (cf. Dingwell and Webb 1990) or if the strain rate locally increases
(Coats et al., 2018; Lavallée et al., 2013; Lavallée et al., 2008).
Nakada and Motomura (1999) proposed that faulting of this spine formed due to a lower
effusion rate that resulted in more complete degassing and crystallisation that increased the magma
viscosity. We advance that fluctuations in pore pressure (Farquharson et al., 2016a) and local strain
rates (Coats et al., 2018; Lavallée et al., 2013; Wadsworth et al., 2019) may be especially important in
triggering embrittlement of otherwise ductile magma. In the ductile regime, strain is accommodated
over prolonged duration without necessarily leading to any substantial stress drop (Coats et al., 2018).
Thus, under such conditions, we do not expect to detect any, or much, seismicity that would characterise
magma rupture near the conduit margin (e.g., Neuberg et al., 2006; Thomas and Neuberg, 2012;
Kendrick et al., 2014b). As a result, we anticipate that magma shearing below the point of rupture (ca.
0.5 km at Unzen; Umakoshi et al., 2008) would have compacted and partially shut the permeability of
the conduit margin, with the shear zone creating an impermeable barrier preventing gas from escaping
to the surrounding country rock and promoting outgassing through the more permeable conduit core, at
least up to the point of rupture (cf. Collinson and Neuberg, 2012). Upon further ascent, changes in the
stress fields and physical properties of the magmas during pulsatory ascent would have favoured
transition to a macroscopically brittle response to shear (Lavallée and Kendrick, 2020), triggering
seismic rupture (Umakoshi et al., 2008; Lamb et al., 2015) and initiation of predominantly fault-
controlled, stick-slip dynamics in the final stint of magma ascent and spine extrusion (Hornby et al.,
2015). In brief periods of high discharge rate, shear may have localised along the primary seismogenic
fault, simultaneously creating a Riedel fracture, but in periods with lower discharge rates, shear would
have been distributed over a wide area and the fault would become inactive (stick phase), shifting the
Riedel fracture to shallower depth; upon renewed discharge rate increase, shear would narrow again,
and faulting would generate another Riedel fracture, and so on (Fig. 9a). Indeed, using seismic events
as a proxy for the ductile-brittle transition it was possible to identify its migration through time as the
inclined spine loaded and compacted its lower shear zone as it grew, dilating the upper fault zone (Lamb
et al., 2015). This is further indicated by the localisation of fumaroles along the upper spine margin
(also observed during our latest field campaign in 2016), showing that the fault zone around the inclined
spine controlled fluid circulation in the upper conduit (Lamb et al., 2015; Yamasato, 1998). Finally, a
late lateral shift in dilational shearing, from the conduit margin to the conduit core, suggest that the
location of shear may migrate during magma ascent in conduits as a result of changes in local stresses
(e.g., upon extrusion and/ or blocks calving), likely resulting from a combination of pore pressure
fluctuations, strain rate reduction and progressive inward cooling which would have favoured
deformation in the core of the spine. Thus, the rheology of magma and the dominant shearing mode
may evolve during ascent, which in turn dynamically modifies the permeability distribution across the
conduit through time (Fig. 9a).

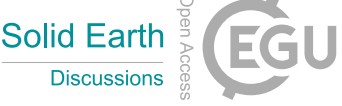

*Rheological assessment of magma switching from ductile to brittle deformation*
The above rheological description is primarily based on the unavoidable decompression of erupting
magma (which degases, crystallises and viscously stiffens), yet previous observations at Unzen suggest
that the conditions for magmatic flow may have fluctuated (Umakoshi et al., 2008; Lamb et al., 2015),
thus contributing to rheological shifts. Here, we invoke findings from the literature to assess the
conditions leading to rupture. The discharge rates associated with spine extrusion in 1994-95 varied,
although Yamashina et al. (1999) constrained a relatively constant spine protrusion rate of 0.8 m d$^{-1}$
over a week-long period in early November 1994. Scrutinising within this period, however, seismicity
indicated a pulsatory magma ascent in the conduit at shorter timescales (Umakoshi et al. 2008; Lamb
et al. 2015). In particular, waveform correlation of the seismic record performed by Lamb et al. (2015)
revealed rhythmic seismicity punctuated by two primary clusters that were attributed to recurring
rupture associated with stick-slip cycles. They identified 668 repetitive events over the course of the 36
days examined: 487 from cluster 1 and 181 from cluster 2. Progressive shallowing of cluster 1 source
location was argued to result from progressive compaction of the lower shear zone (underneath the
inclined magma column) as eruption slowly waned; in contrast, cluster 2, which was accompanied by
low-frequency coda associated with fluid resonance, showed deepening of source location due to
dilation on the overside of the inclined conduit. Considering the events in cluster 1, we define the
recurrence rate of fault slip at 13.5 events per day; so each 'stick' interval for viscous flow would have
lasted on average 106 minutes. Hornby et al. (2015) statistically analysed the slip duration of seismic
events in clusters 1 and 2, defining a mode and mean of 0.1 s. In order to pursue a quantitative analysis
of stick-slip behaviour, we must first turn our attention to our knowledge of Unzen magma flow and
failure conditions.
Coats et al. (2018) studied the rheology of Unzen's porous lavas to define a failure criterion.
Considering the estimated eruptive temperature of ca. 870-900 °C (Holtz et al., 2005; Venezky and
Rutherford, 1999) and measured glass transition temperature (at 10 °C min$^{-1}$) of 790 °C (Wallace et al.,
2019), Coats et al. (2018) empirically defined that Unzen magma would break if experiencing strain
rates exceeding $\sim 10^{-3}$ s$^{-1}$; otherwise, magma would undergo ductile flow. But these determination were
done at atmospheric pressure, so the melt was considered dry; Kusakabe et al. (1999) determined the
concentration of magmatic water dissolved in the groundmass glass of eruptive products at 0.1-0.5 wt.
%; however, the concentration of dissolved water at the point of rupture, at 500 m depth or $\sim 10$ MPa
pressure considering a nominal rock density of $\sim 2,000$ kg m$^{-3}$ (Scheu et al., 2006), would have been $\sim 1$
wt. % (Liu et al., 2005). Such a higher concentration would lower the viscosity of the interstitial melt
one order of magnitude; as the strain rate limit shares an inverse relationship with viscosity (e.g.,
Dingwell and Webb, 1989), we advance that the presence of dissolved water in the melt would have
shifted the strain rate limit by approximately one order of magnitude. If we omit any upscaling of the
above failure conditions for simplification and assume that deformation was localised in the $\sim 1$ m-wide
high shear area of the spine, rupture would have occurred when the ascent rate exceeded 1 mm.s$^{-1}$. As
such high deformation rate episodes are inferred to have triggered fault slip events lasting on average
0.1 s (Hornby et al. 2015), each slip event may have resulted in a mere $\gtrsim 0.1$ mm of displacement. With
13.5 events per day, this would culminate in $\gtrsim 1.35$ mm of magma ascent ascribed to faulting activity,
signifying that deformation associated with the $\sim 0.8$ m daily ascent was predominantly ductile and
aseismic.
We can then turn our attention to geometrical constraints from our structural analysis to frame magma
ascent conditions that satisfy the above failure criterion. The Riedel fractures that are observed at regular
intervals of $\sim 4.5$ cm in the high shear zones have been shown to be important stress and strain rate
distribution markers in multiphase materials containing a weak phase, such as melt and bubbles (Finch



et al., 2020), and can thus be used to constrain rates. Considering the ephemeral nature of Riedel fracture
development (Finch et al., 2020), here we assume that their formation may be encouraged during brief
periods of high strain rate, and they thus portray the clockwork ticking of seismogenic slip events during
magma ascent. Bearing in mind an average spacing of 4.5 cm and an angle of 57° with respect to the
main C-S fabric, we estimate the offset of the loci of rupture events at 5.4 cm. Recalling the 0.1 mm of
displacement ascribed to faulting events (detailed in the previous paragraph), this suggests that ductile
deformation was responsible for 5.3 cm of magma ascent during inter-seismic periods (i.e., inter-
seismicity deformation, ISD; Fig 9a). Again, considering shear over 1 m area and inter-seismic periods
of 106 minutes, we estimate that ductile deformation would have proceeded at an average rate of $8\times10^{-6}$ $s^{-1}$; a value well within the ductile regime as experimentally constrained by Coats et al. (2018). The
above rates (of magma flow in the ductile regime and of faulting) may be conservative estimates,
especially if we consider the rheological consequences of dissolved water at depth. Even if the threshold
strain rate for seismogenic faulting were an order of magnitude higher, at $10^{-2}$ $s^{-1}$, this would only require
13.5 mm of magma ascent in each brittle faulting event and that inter-seismic periods of ductile
deformation at a rate of $\sim8\times10^{-5}$ $s^{-1}$ would have dominated spine extrusion.
In concert the physical and structural description bolstered by the rheological analysis argue for
changes in magma rheology during decompression and pulsatory ascent. We propose that throughout
its journey to the Earth's surface, magma may undergo several cycles of expansion (from vesiculation
and dilation) and collapse (from outgassing and compaction) due to variable permeability and pore
pressure, which may promote switches in shearing regimes that trigger further changes in the
permeability structure of shallow conduits. For instance, the vesicles of low permeability magma may
accumulate fluid, thus reducing the effective pressure and promoting brittle, dilatant rupture; rupture
would in turn allow magma outgassing and a reduction in effective pressure, promoting compaction
and lowering of permeability; and the cycle may recur. The picture portrayed here highlights the need
to understand the coupling between magma and fluid flow dynamics and, importantly, pressure
fluctuations (Michaut et al., 2013) in volcanic conduits with increased spatial and temporal complexities
in order to resolve the transient state of magma and reconcile gas emission data and volcanic eruption
style (Edmonds and Herd, 2007).

**6.    Conclusions**
The present detailed study of the Mount Unzen spine reveals the competing occurrence of
compactional and dilational shear regimes during magma ascent in volcanic conduits. At depth, in areas
subjected to high effective pressure, shearing may induce pore compaction, thereby lowering the
permeability of the system and inhibiting lateral outgassing to the country rock. At shallower depth,
where the effective pressure may be low, shearing may favour localised dilation that enhances
permeability. Both shear regimes result in the development of permeability anisotropy, with
permeability generally being highest parallel or sub-parallel to the direction of extrusion, and lowest
perpendicular to the shear plane. The observation of shearing mode overprints suggests that fluctuations
in effective pressure and strain rates, during stick-slip cycles, may result in magma switching between
compactant and dilational shearing regimes, thus dynamically reshaping fluid circulation at a range of
scales, and in turn controlling outgassing efficiency during magma ascent and eruption.

**Acknowledgements**





We are thankful to Guðjón Eggertsson for help with the maintenance of the permeameter. This project
was financially supported by a European Research Council (ERC) Starting Grant on Strain Localisation
in Magma (SLiM, No. 306488) and an award from the DAIWA Anglo-Japanese Foundation (grant No.
11000/11740). YL and JEK acknowledge support from the Leverhulme Trust (ECF-2016-325 and RF-
2019-526\4, respectively). HT was supported by a University Research Fellowship from the Royal
Society.

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

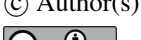

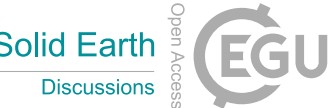

**Figure Caption**
Figure 1. a) © Google Earth image showing the location of Unzen volcano on the island of Kyushu,
Japan. b) Photograph of Unzen volcano, looking northwest, viewed from near Onokoba in the suburbs
of Shimabara city. c)Photo of the relict 1994–95 spine at Unzen volcano (looking westward), showing
(I) the central shear zone (i.e., the cavitation structures detailed in Smith *et al.*, 2001, further expanded
in the inset); (II) the marginal shear zone, bordered by a fault (dark orange-brown colour), and (III) a
large block of sintered breccia of earlier domes, which has become welded to the fault material and
extruded with the spine. Adapted from Hornby *et al.* (2015). d) Photograph of a fragment of the spine
showing the primary internal structure of the shear zone, bordered by a set of closely spaced, inclined
fractures to the left and indurated breccia to the right.

Figure 2. Location of the lava spine blocks and characteristics of the marginal shear zone. a) An aerial
view of Unzen lava dome summit showing the remnants of the 1994-95 lava spine, including the main
spine, the central shear zone (CSZ) block and the marginal shear zone (MSZ) block; image taken from
© Google Earth. b) Photograph of the main spine inclined towards the east. c) 3D construction of the
marginal shear zone block (created using the photogrammetry 3DF Zephyr by 3Dflow). The outcrop
is annotated to show the location of samples (A-H) as well as the 4 main regions (gouge as well as
high-, moderate- and low-shear zones) and key features, including the fault contact (red dashed
curve), shear zone transitions (yellow dashed curves), extension of tensile fracture (C; green lines)
and Riedel fractures (blue curves). The inset shows detail of the fault plane, dividing the gouge and
high-shear zone. Directional arrows X, Y and Z show the orientation of sample coring relative to the
shear plane. d) View of the MSZ block parallel to the shear plane and perpendicular to the shear
plane. Insets show surface textures across the shear zone.

Figure 3. Composite figure of the microtextural characteristics across the marginal shear zone
consisting of photograph of fresh surface textures, plane polarised light (PPL) photomicrographs,
ultraviolet (UV) light photomicrographs and backscattered electron (BSE) images of the groundmass.
Images of the fresh surface were taken following cutting the sample perpendicular to shear.
Phenocryst observed include plagioclase (P), amphibole (A), biotite (B) and quartz (Q). Green boxes
on PPL photomicrographs show the location of the UV light images, which highlight the pore
structures across the MSZ. On UV light images, two white arrows pointing away from each other
show the location of fractures within the groundmass (samples G and H), single arrows point to large
pores adjacent to large phenocryst (samples G and H), and two arrows pointing towards each other
show compaction bands (their spacing represents the width of each band; samples B and C).

Figure 4. Tomographic reconstructions of four samples across the shear zones: a-b) A, c-d) C, e-f) E,
g-h) H.; The upper row shows density-based images of tomographic reconstructions, whereas the
lower row highlights the porous network in blue and the solid fraction is transparent. The
reconstruction shows that the porous fraction becomes increasingly localised towards the fault plane
(i.e., from right to left).

Figure 5. a) Porosity and permeability (parallel and perpendicular to shear plane) profile across the
shear zone, showing the compactant (ductile) nature of the high shear zone, overprint by localised,



dilational (brittle) fractures. Measurements on the gouge sample are plotted at a distance of 0 m. b)
Porosity reduction as a function of effective pressure, derived from the volume of water expelled
during loading in effective pressure of samples cored parallel to shear. Note that the initial porosity
value (at Peff ≃5 MPa) is that of the sample initial porosity (before loading); the exact quantity of
volume expelled between 0.1 and 5MPa cannot be accurately determined due to the method used,
hence we simply show the porosity reduction from this point onward.

Figure 6. Permeability of the marginal shear zone as a function of effective pressure and direction to
shear: measurements conducted a) parallel and b) perpendicular to the shear plane. The data shows a
reduction in permeability with effective pressure; yet the permeability profile across the shear zone
remains, irrespective of the pressure conditions tested. The data shows contrasting permeabilities as a
function of direction, which create c) permeability anisotropy, cast here as the ratio between the
permeability parallel and perpendicular to the shar plane. The anisotropy is most pronounced in the
high shear zone and generally increases as samples were loaded to higher effective pressure due to
fracture closure. Note that the x-axis was truncated and the scale was expanded for the near-fault
high-shear zone for which we conducted more measurements due to the structurally complex nature
of this area of the spine. Measurements on the gouge sample are plotted at a distance of 0 m.

Figure 7. a) Photograph showing measurement locations for the field-based permeability
measurements, for the upper (orange) and lower (green) transects. b) Permeability data for the upper
(orange) and lower (green) transects, plotted against distance. The data shows a drastic increase in
permeability of ~3 orders of magnitude.

Figure 8. Permeability-porosity relationship for Unzen dome lavas and similar effusive lavas. Blue
and red circles represent data from this study, made parallel and perpendicular to the plane of shear,
respectively. Grey circles show porosity data for Unzen from Mueller *et al.* (2005) and Kendrick *et al.*
(2021), and open circles show permeability measurement on USDP drill cores from Watanabe *et al.*
(2008). Other symbols show data for effusive products at similar dome eruptions.

Figure 9. a) Conceptual model showing rheological shifts and evolution of permeability (seen as fluid
flow vectors) during pulsatory magma ascent and stick-slip faulting. The sketches illustrate the
evolution of the extent of active shear zones (in orange), inactive areas (dark reds), active faults (blue)
and inactive faults (grey), during magma discharge fluctuations. The dominant rheology in each area
is numbered (1-4) and is linked to the deformation mechanism map for magma (shown in b). The
sketches (a) show that shear narrows toward the eruption point as magmas is subjected to lower
effective pressure (as shown in b). Compaction of the outer margin of the shear zones (dark red-
brown) would generate a zone of lower permeability (which may act as a local fluid flow barrier) As
discharge rates increase, the width of the shear zone also narrows, and promote a switch to brittle
failure at shallow depth (~500 m), causing the propagation of a primary fault plane and an adjacent
Riedel fracture (which channels fluid flow; blue arrow). Upon discharge rate reduction, the shear zone
would widen again and the fault would become inactive (stick phase), shifting the Riedel fracture to
shallower depth. Upon renewed discharge rate increase, shear would narrow again, and faulting would
generate another Riedel fracture. Thus, the distance between Riedel fractures may be used to resolve
the magma ascent associated with inter-seismicity deformation (ISD). b) Deformation mechanism



map for magma (adapted from Lavallée and Kendrick, 2020). Yield caps are displayed by blue and
green lines representing brittle rupture (dilatant shear) and ductile cataclastic flow (compactant shear),
respectively, and showing an increase in strength as a function of strain rate ($\dot{\varepsilon}$). At low strain rates or
at high effective mean stress, magma flow viscously. The numbers refer to scenarios as displayed for
different parts of the magmatic column in panel (a).




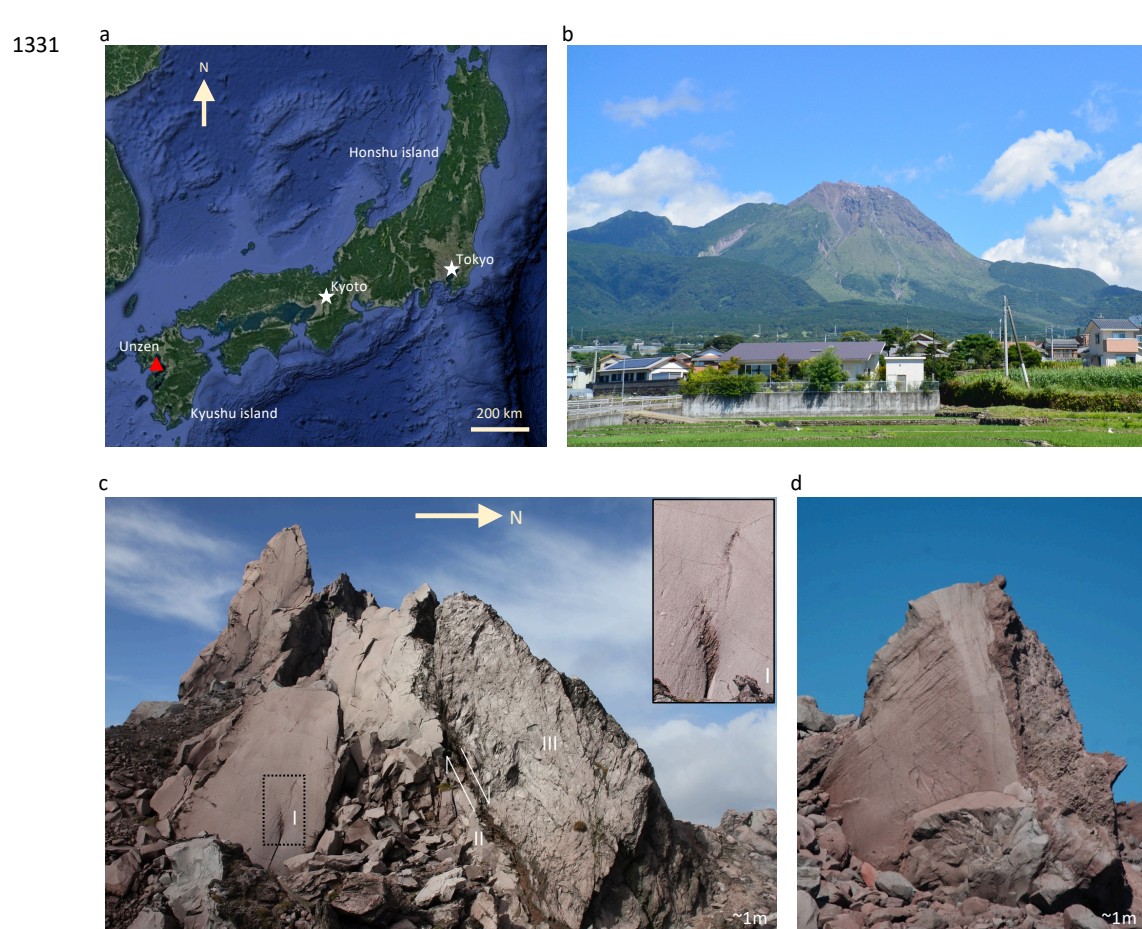

Figure 1.





Figure 2.



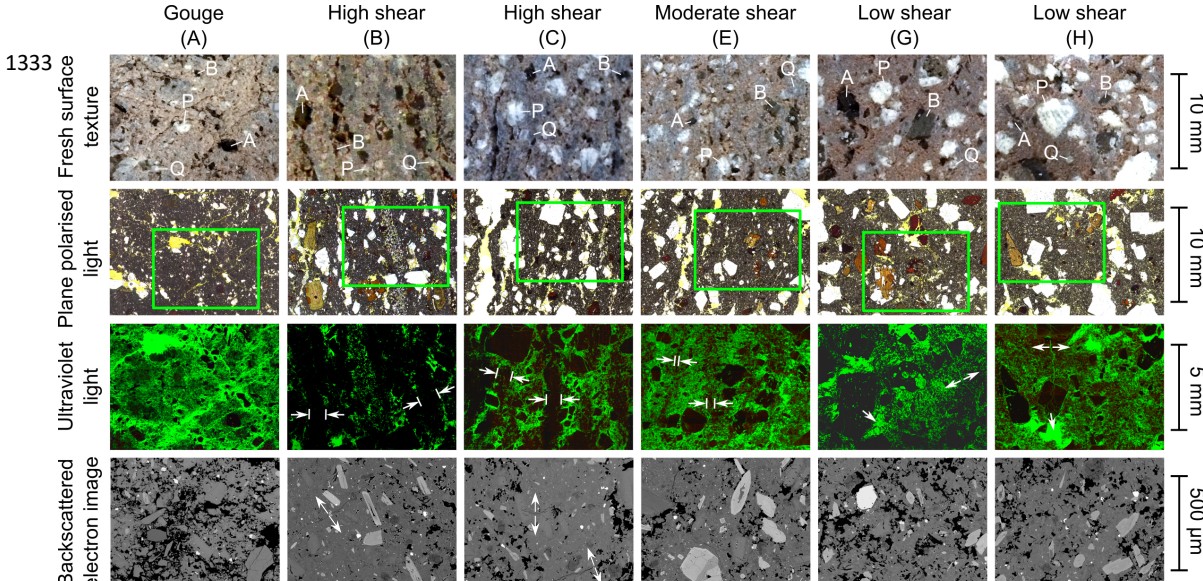

Figure 3.



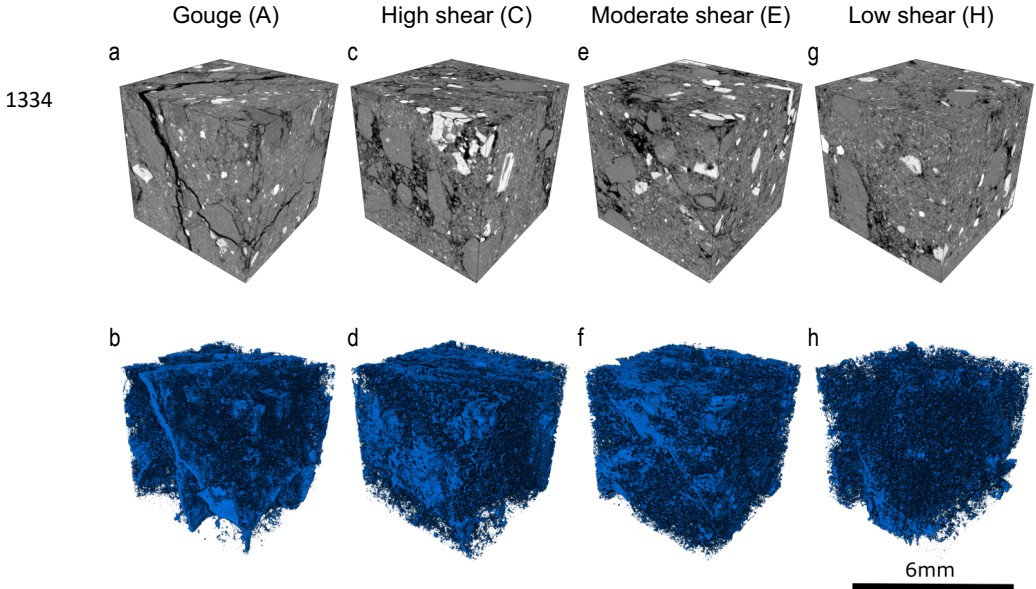

Figure 4.



a


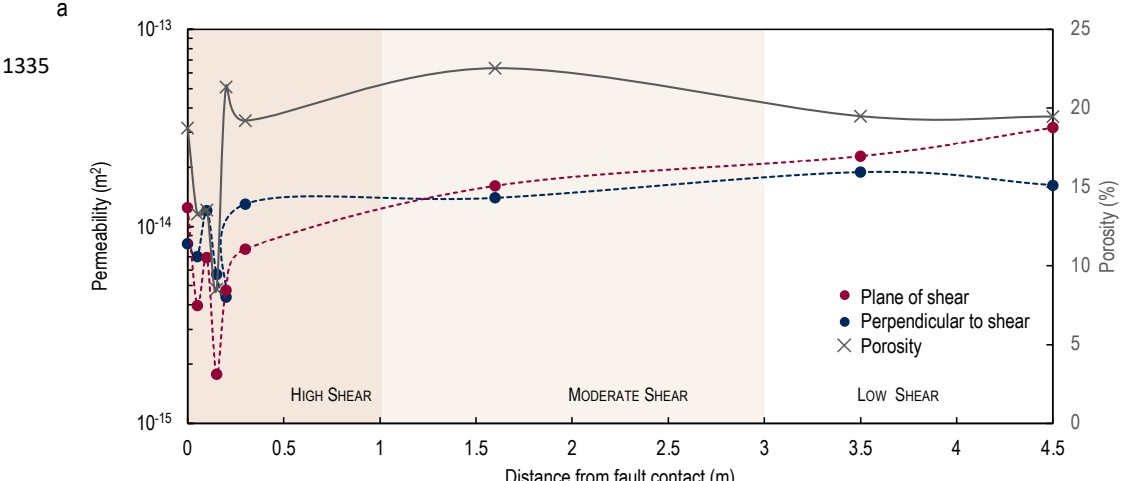

b

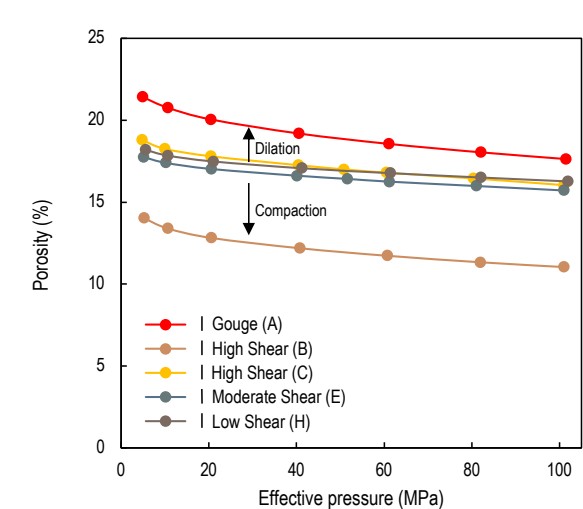

Figure 5.






Figure 6.






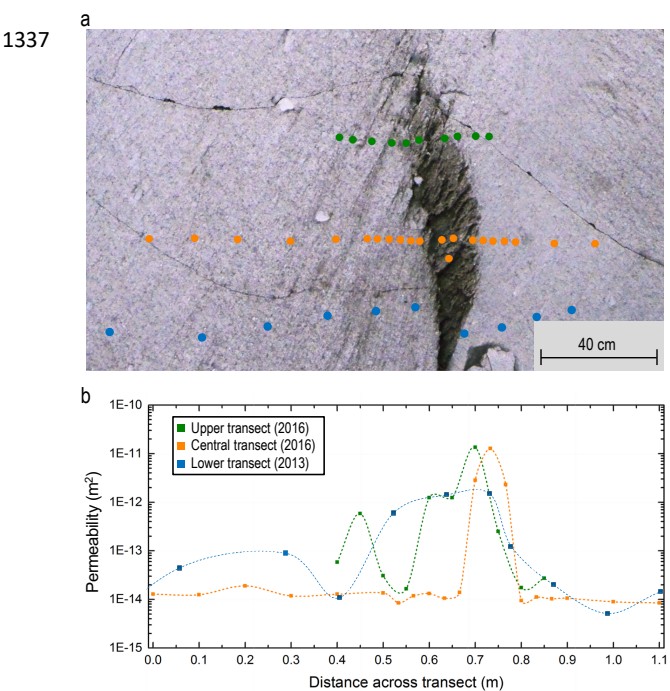

Figure 7.






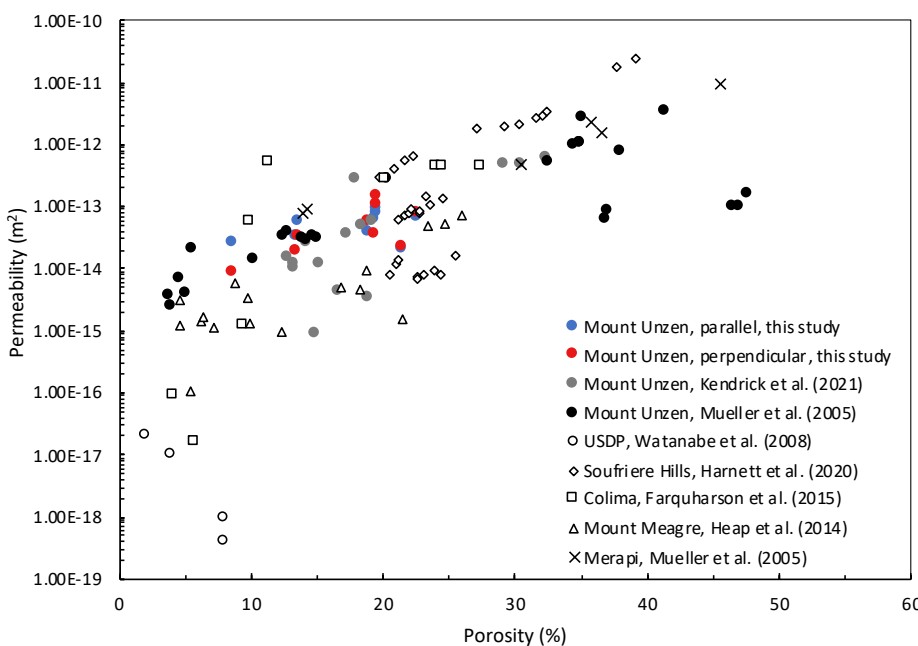

Figure 8.




Figure 9.