# Peer review of "Transient conduit permeability controlled by a shift between compactant shear and dilatant"

_Solid Earth, 2021_

## Author Comment (AC1)

Dear Dr. Heap and Dr. Longo

We are pleased to provide you with replies to the comment made on our manuscript entitled "Transient conduit permeability controlled by a shift between compactant shear and dilatant rupture at Unzen volcano (Japan)". The comments were very good and constructive and raised important inconsistencies which have been addressed and corrected.

Before going any further, we would like to thank you, respectively, for the review and the editorial handling of our article, and for your patience with the delays in the preceding weeks. Three of the leading authors (including myself) are currently in the process of moving from our institution, and three co-authors are no longer in academia, hence it has been challenging to coordinate the response in time. We thank you for the generous extension.

Below you'll find a detailed response (in green) to the comments (in black). The studies cited to support our responses are all listed in the main text.

**Reviewer 1 – Mike Heap – 20 Oct. 2021**

This manuscript presents a study that describes (using field observations, microstructural work, and laboratory measurements) conduit shear zones at Unzen volcano in Japan. The manuscript contains observations and data that will surely interest the volcanological community. I have five main comments and a series of line-by-line comments.

Mike Heap (University of Strasbourg, France)

Main comments

1. When discussing the textures/microstructure of the samples, what is described in the text is often difficult/impossible to see in the accompanying figures. Instances that require attention are described in the line-by-line comments below.

It is indeed challenging at times to clearly show structures, naturally preserved at a range of scales. The comment also made us realise that Figure 3 was too small to clearly show certain features; as such, we've opted to rotate it and enlarge it, to ensure it occupies a wider area of a printed page, easing observation of the smallest microstructures mentioned in the text. We'll further address our response below, in the line-by-line comment section.

2. The XCT data are used sparingly, and qualitatively. It is possible, using the Avizo software used by the authors, to provide more quantitative analyses to support the statements in the text that refer to the size, shape, and connectivity of the void space. Such analysis would, in my opinion, elevate the manuscript. This is discussed in more detail in the line-by-line comments below.

The study is centred on permeability measurements and an attempt at linking micro- and macro-structures to resolve magma evolution leading to eruption. Indeed, it would be possible to quantify the dimensions of certain micro-structures and we pondered it; however, the scanned volumes of each of the irregularly-shaped samples were slightly different (though we crop them to the same volume for the figure) and therefore the resolution of the scans differ and quantitative comparison of size, shape and connectivity of void space would be impacted by this. Moreover, the size of the samples imaged in these reconstructions (only approximately 5 mm across to maximise resolution) show a majority of features which extend beyond the region imaged. Hence, it would be challenging, and arguably impossible, to convincingly interpret the scale of features meaningfully, and extrapolate to quantitatively constrain and conditions of magma ascent. Instead, we chose to show them to help the reader visualise the fabrics at micro-scale.

3. The authors state that the bulk of the one of their conduit shear zones is the result of ductile deformation. Since the entire zone is an impressive manifestation of brittle deformation (it's a shear zone containing cataclasites and fault gouge), I find this statement, at least the way it is written, a little confusing.

We now clarified the text to indicate that indeed, volumetrically, the shear zone is dominantly composed of magma having undergone ductile deformation (indicated by the aligned pores and crystals over a wide zone), whilst the fault margin is thinner and composed of gouge (see fig. 2d for the study area, and for the main spine fig. 1c-d).

I'm also left confused by the discussion about diktytaxitic textures, a texture commonly observed in lavas that contain little to no evidence of shear (this texture can be the result of late-stage gas filter pressing of a silica-rich melt phase, which leaves behind a microlite scaffolding), and ductile deformation. Another way to look at this – as discussed in the recent paper of Ryan et al. (2020, JVGR) that describes a very similar structure – is that large-scale brittle deformation resulted in the formation of fault rocks (cataclasites and fault gouge) that were subsequently lithified by solid-state sintering, and then re-sheared and re-fractured. Did the authors observe any evidence for sintering in their samples? Are the diktytaxitic textures discussed actually sintering textures? I don't want to impose a certain school-of-thought onto the authors, as it's their manuscript, but I consider that sintering-fracturing cycles to be a simpler way to explain the observations/results. Further, the ductile deformation of porous lavas very often results in the formation of compaction bands. If there is ductile deformation, driven by cataclastic pore collapse, where are the compaction bands? Did the authors observe anything like this? Finally, ductile behaviour, driven by cataclastic shear-enhanced compaction, in a very stiff/crystallised magma with a porosity of about 20% would require an effective pressure of about 30 MPa, or depth greater than 1 km. Is this in line with the authors' interpretation? Something to think about.

A diktytaxitic texture describes the irregular, ragged, angular shape of vesicles (or pores in sintered fragmental rocks) influenced by the presence of abundant crystals. They are a descriptive terms observed in a variety of mafic and silicic lavas, possibly produced via different mechanisms. They are commonly observed in crystal-rich lavas, in which exsolved gas preferentially forces its way through interstitial melt leaving vesicle edges impacted by the protrusion of crystals. Previous interpretations have linked such textures, when coupled with

precipitation of cristobalite, to the presence of corrosive fluids, although in our samples no fish-scale cristobalite textures were observed. However, as pointed out by the reviewer, Dr. Heap, more recent work has suggested they can also be formed via solid-state sintering. In the spine erupted at Unzen, we saw no clear evidence for sintering within the samples of the studied shear zone. We merely labelled the observed vesicle-groundmass contact as exhibiting a diktytaxitic texture. Such textures have been observed in several dome lavas e.g., Cordon Calles, Santiaguito, Mount St Helens. In the case of Unzen spine, if the diktytaxitic texture were the result of fault rock or gouge sintering, we would expect the texture to be localised/exclusive to the gouge (A) and high shear zones (B-C), yet they are observed across the entire spine and even some non-spine dome rocks. We agree that shear is not required for the textures to form; yet, we concur that the text in the Interpretation section may not have distinguished between he texture and shear sufficiently. Hence, we rephrased this section of the interpretation to:

> "Smith et al. (2001) invoked the effects of gravitational forces during post-emplacement flow of the lobes as a mechanism for the development of 'ragged' pores and porous/dense flow banding in dome lavas at Unzen volcano. Yet, such diktytaxitic structures have been observed in small surficial dome blocks at Santiaguito volcano (Guatemala), which have not suffered from gravitational effects associated with flow along the flanks (Rhodes et al., 2018); they have also been observed at Merapi volcano, where they were attributed to late-stage gas filter pressing of a silica-rich melt phase (Kushnir et al., 2016). The commonality between these observations is that they occur in crystal-rich magmas, where crystals hamper the presence and distribution of exsolved fluids and interstitial melt, leading to ragged pores with protruding crystals. At Unzen, the character and distribution of the porous network rather evidence the importance of deformation which was pervasive and commonly compactant across the shear zone. Experiments have shown that in the ductile field, material may deform by sustaining substantial compaction without the propensity for developing localised strain (Rutter, 1986) – a regime that generally results in a permeability reduction through shear (Ashwell et al., 2015; Kushnir et al., 2017b; Heap et al., 2015a; Heap et al., 2015b)."

Regarding compaction bands, they are indeed rather dense and best observable at mesoscale on the outcrop. In thin section, it is difficult to visualise them clearly as their textural scale approaches that of the thin section, making them hard to scrutinise clearly.

As for the inference for the depth for ductile behaviour, here we invoke the likely occurrence of both, viscous flow and shear-enhanced compaction. Viscous flow would have taken place at any depth in inter-seismic periods; in contrast, shear-enhanced compaction would have likely only happened below a certain depth. As we do not hold an experimental constraint coining the conditions for shear-enhanced compaction in crystal-rich porous magma, here we revert to seismic analysis which suggests that faulting took place at ~500m depth (e.g., Umakoshi et al., 2008; Lamb et al., 2015); hence we infer that shear-enhanced compaction may have taken place at greater depth, as the rupture events which formed the gouge cross cuts the shear bands. As such, the pressure of 30 MPa (and so depth) suggested by the reviewer, based on rock properties, aligns with our argument that it occurs prior to ascending to 500m.

4. Similar observations/data exist for Mt St Helens (Gaunt et al., 2014, Geology) and Chaos Crags (Ryan et al., 2020, JVGR). It would be interesting, in my opinion, to compare and contrast the various systems somewhere in the discussion section. Are the permeabilities similar? Is the evolution of permeability as a function of distance from the fault/margin similar?

Good point. we have now included some sentences in the discussion, comparing these studies.

5. There are various aspects of Figure 9b that I disagree with. First, I'm not sure why the authors have added "high pore pressure" and "low pore pressure" to the x-axis. The effective mean stress is a term that neatly encapsulates the three compressive principal stresses and the pore fluid pressure (i.e. a low effective mean stress doesn't necessarily mean that the pore pressure is high) and so it not needed/incorrect to refer to the pore pressure on the x-axis. Second, the authors have also added "low strain rate" and "high strain rate" to the y-axis, effectively giving the diagram two y-axes.(…)

We agree with these statements. The terms "low pore pressure" and "high pore pressure" as well as "low strain rate" and "high strain rate" had simply been added in a figurative sense as "comments" to somewhat illustrate the type of conditions which may control the position in this figure. For instance, it is not a simple requirement for the principal external stresses to be high in order to reach a high effective mean stress, as a low pore fluid pressure could lead to the same result at a low confining pressure. But indeed, we appreciate that the statement are not strictly true at all time and so, adds confusion. As such, the mention of pore pressure and strain rate was removed from the axes and moved into the figure next to the arrow pointing in the direction in which different conditions may modify the position in this diagram.

(…)However, a high differential stress does not necessarily mean high strain rate, and vice-versa.(…)

Reading this comment, we realise that some of the annotation in the diagrams may not have been clear, perhaps initiated by the labelling of the Y axis with the term "strain rate". Yet, it can be reasoned that for a given effective mean stress an increase in differential stress will always increase the strain rate. It is however also true that for a given differential stress, both an increase and a decrease in effective pressure may increase the strain rate; that is respectively the case in the ductile and brittle fields. It is a concept difficult to represent on the graph and we realise that the sole, grey, upward arrow we showed in the figure created confusion. In fact, the way to read this figure is by following lines, as the lines represent iso-strain-rates; the blue, green and red lines reflect the range of conditions that promote a given strain rate in each regime. Considering this point, we now show two inclined arrows to represent the impact of strain rate on shifting the position of the iso-strain-rates lines in the ductile and brittle regimes; equally, the caption has been modified to explain what the lines are and why strain rate would impact the conditions differently in each deformation mode.

(…) As a result, the graph shows that, at high strain rate and high effective mean stress, the material will be ductile. However, magma will likely be brittle at high strain rate, no? (…)

And that is indeed what the graph shows. The schematic contains two ductile to brittle transitions: the viscous to brittle transition (long-dash line) and the cataclastic to brittle transition (short-dash line). Both of which exhibit a positive slope, indicating that as strain rate increases, the brittle field is met. In the case of the cataclastic to brittle transition, it would be met at very high strain rate if the effective mean stress is very high. Moreover, the graph shows that at certain effective mean stresses, an increase in strain rate, can force a magma, otherwise flowing viscously, to flow cataclastically; then, further increase in strain rates could further switch the deformation to the brittle field. We believe this point has been dealt with by the changes to the figure caption which now explains how to read along the lines on the diagram.

(…) Similarly, at a high effective mean stress, the deformation mechanism can be changed from viscous flow to cataclastic flow by increasing the differential stress. (…)

That is indeed correct and illustrated in the sketch: This transition is represented by the dotted line.

(…) Why would increasing the differential stress change the micromechanism of deformation at a constant effective mean stress? (…)

It does so for the same reasons that as strain rate increases for a silicate melt it will meet the glass transition and rupture. At a given high effective mean stress, an increase in differential stress would lead to higher strain rate, thus prompting rupture at the glass transition. In this case, however (as we consider a porous magma), that would translate into a scenario in which the melt pervasively fractures between vesicles, and in turn crushing vesicles, leading to cataclastic flow.

(…) Increasing the differential stress at low effective mean stress can change the behaviour from viscous flow to shear rupture, and increasing the effective mean stress can also change the behaviour from viscous flow to shear rupture. In other words, increasing the confining pressure could change the behaviour from viscous flow to shear rupture. I don't see how this can be true. (…)

Here we only partly concur with the comment made. Indeed, "increasing the differential stress at low effective mean stress can change the behaviour from viscous flow to shear rupture" (as detailed above); the transition between the two is shown by the long-dash line. However, "increasing the effective mean stress can also change the behaviour from viscous flow to shear rupture" is not a concept that has been experimentally detailed and there is no evidence for this. Instead, we find that with an increase in effective mean stress, the deformation of a porous melt may rather switch from viscous flow to cataclastic flow (as shown in the sketch). Here we'll reason through this diagram, looking at the effects of confining pressure and pore pressure individually to argue this point. If one takes a porous magma and increases the confining pressure (without changing the differential stress or the pore pressure), then strain rate increases and the porous structure would be crushed, leading to cataclastic flow. The same would be true if one bleeds the pore pressure out of a bubbly magma, as the vesicles would be crushed by the increasing external forces, causing high strain rates that prompt distributed fractures, leading to cataclastic flow.

(…)In fact, there are numerous instances in which the diagram suggests an unlikely/incorrect material behaviour for a certain set of x- and y-coordinates. It is not possible on a graph like this to simply exchange the y-axis (differential stress or strain rate) to explain certain aspects of the diagram. (…)

As previously stated, we realise that our earlier indication of low and high strain rates on the Y axis (and low and high pore pressure on the X axis) led to confusion, and made it difficult to read the figure in the way it was intended. We hope that our clarifications helped improved the readability of this schematic (as we now urge the reader to read this diagram by following lines, rather than to pick out specific x- and y- coordinates) to assess the conditions leading to different deformation modes in magma.

(…) Finally, the authors have also indicated the influence of strain rate directly on the diagram, by providing a series of curves. However, while the influence of strain rate on the brittle failure of materials is reasonably well-known, the influence of strain rate on the onset differential stress for cataclastic flow is, to my understanding, largely unexplored. How are the authors sure that increasing the strain rate increases the differential stress for the onset of shear-enhanced compaction? In short, the diagram combines aspects of rock behaviour and fluid behaviour that I don't consider to be compatible. I strongly suggest that the authors reconsider this diagram.

Whilst it is true that many areas of this diagram remain unexplored by laboratory experiments, the general attitudes of the iso-strain-rate lines in each deformation mode, and their intersections, which define key transitions (between viscous/ cataclastic/ brittle deformation), can be reasoned and argued to be appropriate; yet they remain unquantified, hence the diagram makes no mentions of values. This diagram has been built, conceptually, by reviewing and considering all mechanical data obtained on magma as well as concepts which have derived from decades of work in rock physics. It is a task that took a very long time, and which is detailed in other review papers (Lavallée and Kendrick, 2020, 2022), and we fully appreciate that interpreting the content of the diagram requires time and patience. As for its incorporation in the manuscript, we opted to keep it in, with modification and clarification to the caption, as it supports the discussion and helps us discuss the key variables which may have contributed to magma ascent at Unzen. It also puts into context the scenarios in Fig 9a. We hope that the corrections we made to the figure, its caption, and the main text, have helped clarify the message, as we feel it is an addition that helps this study and may help the interpretation of observations in other studies from the community.

Line-by-line comments

Line 118: Another relevant, and very recent, study is that of Ryan et al. (2020, JVGR). These authors document (field observations, microstructural work, laboratory measurements of porosity and permeability) a conduit-parallel shear zone within crystal-rich dacitic magma at Chaos Crags (USA).

Ryan, A. G., Heap, M. J., Russell, J. K., Kennedy, L. A., & Clynne, M. A. (2020). Cyclic shear zone cataclasis and sintering during lava dome extrusion: Insights from Chaos Crags, Lassen Volcanic Center (USA). Journal of Volcanology and Geothermal Research, 401, 106935.

Thank you. The study is very interesting has been cited herein; we took the occasion to reorder certain statement to better present the content.

Lines 195-197: Brittle deformation does not always lead to increases in bulk sample porosity. As shown in the cited Heap et al. (2015, BV) paper, samples that developed macroscopic fractures can contain a lower bulk porosity at the end of the experiment. I would subtle reword this sentence.

Indeed, in some cases, the total porosity may lessen slightly following brittle failure, but it remains that the fracture itself tends to increase porosity locally. We should clarify that the statement did not say that "Brittle deformation always leads to increases in bulk sample porosity"; it rather said "brittle failure results in dilation (i.e. the creation of porosity)", and that, is generally true, at least in the area of the fracture. As such, to prevent potential

misunderstanding, we clarified and reworded of statement to "generally results in local dilation".

Line 207: "strain rates are…".

Corrected

Line 210: Additionally, efficient compaction can shunt the magma into the brittle regime once porosity has been reduced sufficiently to promote brittle behaviour at the imposed effective pressure.

Indeed. This point was added.

Line 237: "…the source of debate…"?

Corrected.

Line 275: CSZ was already defined.

Thanks for spotting this duplication.

Line 288: What was the length, or range of lengths, of these samples?

The length was defined in the next sentence already, but this comment made us notice an inconsistency and so we edited the text to read: "For the largest samples (A, B, C, E, H; see Fig. 2c-d) a set of 2-3 cylindrical cores (two parallel and one perpendicular to shear plane) were prepared with a diameter of 26 mm and a length of 13 to 34 mm, depending on the size constraints of each sample."

Line 321: Distilled or deionised water? Or tap water?

Distilled. This has been added.

Line 323: Since water is expelled from both cracks and pores, I suggest the authors change "pores" to "void space" or "porosity".

Corrected.

Line 325: Did the authors wait at each pressure increment to ensure microstructural equilibrium? I guess the authors waited until the porosity change stabilised before measuring permeability at a given effective pressure?

 Indeed. We have now added the following to the relevant text in the method "Each time the sample was loaded to new confining pressure increment, the volume of water expelled from the void space in a given sample (due to compaction) was monitored to constrain pore volume change due to crack closure as a function of pressure (Lamur et al., 2017); this allowed us to monitor when the samples had equilibrated to the set conditions at each pressure step. Steady-state flow permeability ($k$) was then measured by applying low pore pressure gradients ($\Delta P$) of

0.5 and 1.5 MPa to ensure laminar flow with no slip conditions (Heap et al.2017a) to satisfy Darcy's Law:"

Line 361: Was there any evidence of alteration?

Luckily for us, these blocks (i.e., the last magma to have erupted in early 1995) did not appear to experience post-emplacement alteration beyond mild oxidation due to its slow extrusion, unlike other parts (or lobes) of the dome, erupted earlier, or areas around the main spine which still have active fumaroles.

Line 393 and elsewhere: What is shown in blue on Figure 4 is all connected, or are isolated pores also shown? It is possible using the Avizo software to provide pore size and shape statistics. Why not provide histograms showing pore size distributions for these materials, for example? This type of analysis would use these data more effectively, and provide some quantitative statements to support the authors' claims.

In Figure 4, we show all pore space in blue. The samples used for these reconstructions were not systematically prepared to undertake a quantification of the porous network across the shear zones as was mentioned in reply to the main comment previously. It is simply data we had at hands that helps appreciate, qualitatively, the character of the porous network. To do any quantification, the sample preparation, imaging and analysis would have to be done anew entirely. Hence, we simply keep these images to supplement our observations.

Line 404: It's difficult to observe the irregular vesicles in Figure 4e and 4f. Can the authors isolate one or two of these irregular-shaped pores and show them alongside the reconstruction of the entire sample? The Avizo software could also be used to provide the average circularity of the pores, to provide some quantitative statements to support the authors' claims.

As previously mentioned, we do not intend to quantify the pore space in these reconstructions; this would require a different type of study. The pores in fig. 4e are in black and clearly show that they are irregular. In the lower panel (4f) these irregular (non-spherical) geometries are presented in blue.

Line 405: How are the authors sure that these vesicles "enhance the connectivity"? I'm not sure that changes in connectivity can be assessed using Figure 4.

Because we observe complex vesicular network as well as fractures that traverse from pore to pore, increasing their connectivity, not only in Figure 4 but also in Figure 3, but more fundamentally than that it is generally accepted that fractures enhance connectivity of already porous media (e.g. Tiab and Donaldson, *Petrophysics (4th Ed.)*, Chapter 3 - Porosity and Permeability, 2016). We clarified the text to reflect this.

Line 411: Can the authors label the S- and C- fabrics in Figure 3?

We've added a mention of the S- and C- fabrics in the new Figure 3, which are both parallel in the high shear area. Yet, they are not obvious at this scale; they are more obvious at larger scale in Figure 2 D.

Line 422: It's somewhat difficult to see what the authors are describing in Figure 3. Perhaps it's worth showing a zoomed image showing the pulverised phenocrysts within the more porous bands?

We realised that Figure 3 was too small to enable the reader to observe some of the features we mention in the text. As such, we have rotated and enlarged the figure, adding annotation to highlight more features (e.g., S- & C- fabrics, pulverised crystal) than in the original submission.

Line 432: Unless I'm mistaken, the authors don't refer here to the XCT reconstructions shown in Figure 4c and 4d. What do these data show?

The reviewer is mistaken, Fig 4c and d are presented in this section (3.1.1) at line 418 (in the original submission). They show bands with high porosity, running diagonally across the reconstruction.

Line 444: Did the authors observe any evidence for the sintering of particles in the fault gouge?

No clear evidence was identified; somewhat surprisingly. We did think there would be evidence but at most the gouge is only slightly indurated. Potential reasons for the absence of sintering are multiple; low peraluminous glass and possibly relatively low (near Tg) temperature (promoting sluggish relaxation), low confining stresses in the shallow ~500m following rupture, etc. so we noted neither viscous nor solid-state sintering.

Line 447: It's difficult to see the alignment of pores in Figure 4a and 4b. Can the authors show some additional images that show this alongside the reconstruction of the entire sample?

At the scale shown, 4a-b show primarily, the localised dislocation present in the gouge; the intragranular pores are rather small and better observed in the lower panels of Fig. 3 (column A), particularly in the UV light.

Line 454: I would say "large", rather than "important". It might also be worth noting that these porosities correspond to the porosity at ambient pressure.

We have clarified the text accordingly and settled for "significant" variations in porosity.

Lines 455-446: Although up to the authors, I suggest that they reword this sentence to improve clarity. I had to read the sentence several times to understand its meaning.

Thanks for pointing out the awkwardness of this sentence; we clarified it.

Line 458: Be careful with percentages. Are the authors talking about a relative change or a change in percentage points?

We mean actual changes in porosity, as suggested by using "decrease in porosity of" instead of "decrease in porosity by"; this value is clear when looking at the dataset in Fig. 5b.

Line 467: The permeability values shown in Figure 5 were measured at what effective pressure? 5 MPa?

That is correct. Thanks for pointing out this omission; we've now added a mention of it in the caption.

Lines 481-487: Should some of this text not be moved to the discussion section?

In this section we kept only basic interpretations, intertwined with the observations to maximise the value of the observation – here in order to explain the impact these textures have on the permeability in the context of already established relationships between effective pressure and crack closure – placing this here avoids duplication of information, and ensures that the discussion, which is already bulky, remains focused to keep a linear message across the manuscript. We hope the text is fine in this state.

Line 518: The surface of the fracture is quite rough. Are the authors sure that they had a good seal between the field permeameter and the rock surface? I say this because the permeabilities within the fracture are very high.

Measurements were undertaken several times; however, it is true that the inner surface of the fracture is more even (rougher) than the spine surface. In some instances, measurements were attempted with and without a gum to seal the permeameter to the rock (note that the rubber nozzle of the permeameter is already quite malleable). Both measurements returned the same values so long as we very carefully held the permeameter solidly against the outcrop. We are confident in these values; yet, we appreciate it would have been great to obtain samples for laboratory measurements, but this is not permissible due to its heritage status.

Line 533: Diktytaxitic textures are also documented in dome lavas from Merapi volcano in the cited Kushnir et al. (2016, JVGR) paper. These authors interpreted this texture as the result of late-stage gas filter pressing of a silica-rich melt phase, which left behind a microlite-supported groundmass and cristobalite in neighbouring vesicles.

Good point. Diktytaxitic textures are a petrographic feature which may reflect different petrogenetic processes. Although our list of processes is not exhaustive, we've made mention of the potential roles of gas-filter pressing.

Line 576: "in conjunction"?

Corrected; thank you!

Line 599: Kushnir et al. (2017, GRL) appears in the reference list, but perhaps more relevant, at least here, is Kushnir et al. (2017, EPSL).

Kushnir, A. R., Martel, C., Champallier, R., & Arbaret, L. (2017). In situ confirmation of permeability development in shearing bubble-bearing melts and implications for volcanic outgassing. Earth and Planetary Science Letters, 458, 315-326.

Thanks for pointing out this erroneous Endnote entry. It's now updated.

Line 600: The cited Heap et al. (2017, EPSL) paper also measures porosity changes during the deformation of andesite at high pressure and high temperature (above and below the threshold glass transition temperature at the imposed strain rate).

Good point; this has now been amended.

Line 623: See also the aforementioned paper by Ryan et al. (2020, JVGR).

Indeed; the list of citations here was not exhaustive, as many authors have made this observations. We have now added a few more citations.

Line 647: The authors mean "magmas" rather than "lavas"?

That's right; this has been corrected.

Line 653: The Coats et al. (2018, SE) study only provides uniaxial deformation experiments. This sentence, however, discusses the influence of effective pressure on rock failure mode.

Indeed, thanks for pointing this. The citations used to be merged in an earlier version of this section of the manuscript. We had not corrected the citation accordingly but have now fixed this.

Line 730: "determinations"?

Corrected.

We hope our replies address your questions and any concerns you may have had about the content presented in this study. We thank you for your continued time and efforts on our manuscript, and look forward to your response in due course.

Best wishes from Liverpool,

Yan Lavallée, on behalf of all authors (10 Feb. 2022)

---

## Author Comment (AC2)

Dear Dr. Küppers and Dr. Longo

We are pleased to provide you with replies to the comment made on our manuscript entitled "Transient conduit permeability controlled by a shift between compactant shear and dilatant rupture at Unzen volcano (Japan)". The comments were very good and constructive and raised important inconsistencies which have been addressed and corrected.

Before going any further, we would like to thank you, respectively, for the review and the editorial handling of our article, and for your patience with the delays in the preceding weeks. Three of the leading authors (including myself) are currently in the process of moving from our institution, and three co-authors are no longer in academia, hence it has been challenging to coordinate the response in time. We thank you for the generous extension.

Below you'll find a detailed response (in green) to the comments (in black). The studies cited to support our responses are all listed in the main text.

**Reviewer 2 – Ulrich Küppers – 7 Dec. 2021**

The manuscript presents results from field observations as well as textural and analytical investigations of parts of the final spine on the 1990-1995 dome of Unzen volcano, Japan. This study is well suited for SE and I recommend publication after minor corrections. I have one main consideration and several line-by-line comments.

Ulrich Kueppers, Ludwig-Maximilians-Universität (LMU) Munich, Germany

Main comment: The authors repeatedly invoke "repeated phases of increased magma ascent rate (line 44) or "pulsatory ascent" of magma (766) as underlying reason for the observed repeated brittle-ductile transitions. I would like to see more discussion whether this is considered a source or a path problem, i.e. whether magma ascent velocity varied in the first place or if geometrical conditions of the conduit system and/or the dynamic evolution of the magma may have led to the stick-slip-behaviour.

This is a great point of consideration, which we would appreciate contributing to; unfortunately, this aspect of magma dynamics cannot be resolved with the data presented in our study. This would best be addressed using a modelling approach. We anticipate that there are feedbacks between the source and the response, causing non-linearity in ascent rate; so both may likely impact each other at different times. Here, however, we refrained to enter this debate as our data would not support any claim we could make about it, and instead we rely on geophysical measurements made during the eruption, including repetitive seismicity (see Umakoshi et al., 2008 and Lamb et al., 2015 amongst others), which indicate the pulsatory ascent.

Line-by-line:

26: "in shallow volcanic conduits": not only here, right? and what means shallow? I would delete this vague indication of depth

Indeed; we simply added this caveat as here, we can only comment on shallow systems (e.g., <1 km). Our reasoning was that, as magma viscosity would decrease with depth, we anticipate a depth-dependent transition where permeability may be more important in controlling volatile transfer (i.e., at shallow depths where viscosity is high and bubbles cannot freely migrate, so gas percolates) and a deeper region where gas bubbles can freely migrates through the melt (i.e., not requiring permeable pathways to operate); this transition would be magma specific. Yet, we opted to remove this as unnecessary addition.

32-34: please add some values.

We feel the abstracts contains sufficient detail; any more quantities, would detract from the main point. These values can be accessed in the main text.

46: "partially tore the spine core with slight displacement". unclear, please explain

We rephrased to "which partially tore the core of the spine, leaving a slight permanent displacement."

67-69: no obvious order

This order is pre-set by the Solid Earth template for Endnote. We welcome any further indication for the journal if this should be modified.

352: delete "emplaced"?

Corrected.

396: "under microscopy". better say "under/with a microscope"?

Corrected.

397: please quantify the size of "large vesicles"

We indicated: $\lesssim$ 3 mm

402: can bands "localise"?

Bands are, by definition, localised.

414: "crystal plastic deformation" better change to "plastic crystal deformation"?

We added a dash for clarity as the term is indeed, "crystal-plastic deformation".

419: "few isolated millimetre-size vesicles". how is this possible in a 1 mm wide band (417)

Good point raised here. We re-examined the thin section and revised the size in the text.

444-445: "The fragments in the gouge are generally densely compacted and the porosity is uniformly distributed,". can you comment on whether the compaction took place before mechanical abrasion (rounding) and are a remnant of an earlier texture or if that happenend during sintering?

The gouge does not show sign of sintering; it is merely, poorly indurated. It is hard to tell at what point materials abraded and compacted; presumably, during shear in the final ~500m of ascent. As this took place, the materials would have likely simultaneously compacted, but that remains speculation. We did not add inference on this in the text to refrain from speculating on it.

527: "via crushing of the pore walls". odd wording, please consider rewording

That is common way to describe the textures that form during cataclasis in rock physics.

567: "shallowing of the ascending spine". consider rewording. during magma ascent, the flow field may change such that a plug is formed but a spine is defined as the surficial (= above the dome surface) expression of magma/lava extruded that doesn't change texturally any longer under the acting stress at a given cooling rate.

Thanks for pointing out this inconsistency; we've rephrased to "during shallowing of the magma plug, which impacted fluid flow during spine eruption."

We hope our replies address your questions and any concerns you may have had about the content presented in this study. We thank you for your continued time and efforts on our manuscript, and look forward to your response in due course.

Best wishes from Liverpool,

Yan Lavallée, on behalf of all authors (10 Feb. 2022)